# FROM EVALUATION TO DESIGN: USING POTENTIAL ENERGY SURFACE SMOOTHNESS METRICS TO GUIDE MLIP ARCHITECTURES

## ABSTRACT

The reliability of machine learning interatomic potentials (MLIPs) in downstream physics tasks depends not only on reproducing reference energies and forces, but also on the smoothness of the underlying potential energy surface (PES). While prior work has evaluated smoothness indirectly—most commonly by running microcanonical molecular dynamics (MD) simulations or calculating phonon modes—such tests capture only near-equilibrium smoothness and are computationally expensive. We introduce the Bond Smoothness Characterization Test (BSCT), a simple and inexpensive benchmark that directly quantifies PES smoothness both near- and far-from-equilibrium by probing controlled bond deformations. Since BSCT measures the PES itself, it can detect a wide range of instabilities, such as discontinuities, artificial minima, or spuriously large forces. To investigate how BSCT can guide the design of scalable, physically reliable MLIPs, we start from an unconstrained Swin-Transformer-inspired backbone and conduct a controlled study on the SPICE (molecules) and MPTrj (materials) datasets. Beginning with this baseline, we introduce targeted design changes—differentiable k-nearest neighbor graphs, temperature-controlled attention, and broadened radial smearing widths. At each step, we measure the energy and forces prediction accuracy, energy conservation in microcanonical simulations, and the BSCT metric. Our results show that BSCT improvements consistently predict reductions in MD instabilities and enable early-stage filtering of problematic models. The final BSCT-guided models achieve state-of-the-art accuracy on SPICE and MPTrj while maintaining excellent smoothness, demonstrating that optimizing for physical soundness via BSCT naturally yields high performance. Our results position BSCT as a practical, general-purpose metric for guiding the design of reliable MLIPs.

## 1 INTRODUCTION

Interatomic potentials, which describe the microscopic interactions in atomistic systems, are fundamental to computational chemistry and materials science. Density Functional Theory (DFT) (Kohn & Sham, 1965), one of the most widely used *ab initio* methods, enables essential calculations such as molecular dynamics simulations and geometry optimizations. These calculations support diverse applications, from drug discovery to catalyst design (Cole & Hine, 2016; Jain et al., 2016; Hammer & Nørskov, 2000). However, the computational complexity of DFT scales as $O(n^3)$ with the number of electrons, making it prohibitively expensive for large systems or long timescales.

Machine Learning Interatomic Potentials (MLIPs) have become powerful surrogates for DFT, offering orders-of-magnitude speedups while approaching DFT-level accuracy. However, accuracy on energy and forces predictions alone does not guarantee reliability. Small deviations in predicted forces can accumulate in simulations and relaxations, producing unstable trajectories or unphysical observables (Fu et al., 2022; Bigi et al., 2024; Miret et al., 2025). The underlying issue is often the physical soundness of the potential energy surface (PES).

A critical aspect of physical soundness is PES smoothness. In quantum chemistry, *chemical smoothness* has been rigorously defined as the absence of artificial extrema or inflection points along all directions (Subotnik et al., 2008). This differs from the MLIP community's informal use of "smoothness" to mean bounded PES derivatives (Fu et al., 2025), which often requires costly MD to evaluate.

Smoothness is especially important far-from-equilibrium, where spurious PES features can cause catastrophic simulation failures.

Current MLIP benchmarks provide measures of accuracy and stability: for example, through near-equilibrium test sets (Póta et al., 2024) or observing energy conservation over time in microcanonical NVE simulations (Fu et al., 2025). While these approaches have been valuable to the field, they are either computationally expensive or focus on limited regions of the configuration space. We address this limitation with the Bond Smoothness Characterization Test (BSCT), a benchmark and metric designed to directly and efficiently evaluate PES smoothness in both near- and far-from-equilibrium regimes. BSCT probes one-dimensional bond-stretching and compression scans, where the true PES is known to be smooth, making spurious features easy to detect. Using BSCT, we also define a Force Smoothness Deviation (FSD) metric, which quantifies smoothness at low cost and correlates strongly with MD stability, providing an early indicator of physical reliability without running long simulations.

To demonstrate BSCT's utility for guiding model development, we construct an expressive, unconstrained attention-based backbone and systematically introduce targeted modifications to improve physical soundness and PES smoothness without sacrificing scalability. These include a differentiable k-nearest neighbors (Diff-kNN) algorithm, broadened Gaussian smearing widths, and temperature-controlled attention. BSCT-driven evaluation reveals how each choice affects smoothness, accuracy, and MD stability, yielding models that have strong near-equilibrium accuracy while improving far-from-equilibrium smoothness. Together, BSCT and this model case study provide a practical framework for developing MLIPs that are both accurate and physically sound.

## 2 RELATED WORKS

**MLIP Benchmarks.** Many MLIP benchmarks have been developed to supplement energy and force errors. The TorsionNet-500 dataset includes 500 molecules' torsion scan profiles, allowing MLIPs to compare their PES to DFT calculations (Rai et al., 2022). Fu et al. (2022) proposed to use MD simulation stability and $h(r)$ reconstruction to benchmark models. Bigi et al. (2024) used NVE simulations and Jacobians to measure the non-conservative behavior of the models. Kreiman & Krishnapriyan (2025) explicitly evaluates the generalization capabilities of foundational MLIPs. The NNP Arena provides an assortment of molecular and lattice property prediction benchmarks as well as inference speed test (Rowan Scientific Corporation, 2025). Our benchmark evaluation, BSCT, differs from these by explicitly measuring PES smoothness far from equilibrium.

For materials, the Open Catalyst project assesses models by their ability to predict the correct relaxed energy of the catalyst-adsorbate system (Chanussot et al., 2021). Matbench Discovery ranks models by their ability to predict structure stability (Riebesell et al., 2023). The MDR phonon benchmark tests MLIP's capability to predict the correct phonon structure of the lattice (Póta et al., 2024). MLIP arena provides a wide array of benchmarks for materials ranging from homonuclear diatomics to equations of state (Chiang et al., 2025). CHIPS-FF evaluates MLIPs on predicting material properties such as elastic constants, phonon spectra, defect formation energies, etc. (Wines & Choudhary, 2024).

**MLIPs for Atomistic Systems.** MLIPs can be roughly divided into three categories based on their transformation under the Euclidean group: equivariant, invariant, and no built-in equivariance. Equivariant architectures featurize the embeddings to carry irreducible representation indices. One example of this is Tensor Field Network (Thomas et al., 2018), which was subsequently adopted in NequIP (Batzner et al., 2022). MACE (Batatia et al., 2022) followed a different path to generalize the Atomic Cluster Expansion (ACE) (Drautz, 2019) to a multi-layer message passing neural network. eSCN (Passaro & Zitnick, 2023) and eSEN (Fu et al., 2025) utilize the efficient $SO(2)$ convolution to perform the equivariant graph operations. On the other hand, invariant networks featurize the embeddings to be invariant to the group operations, such that rotations and translations would not change their values. SchNet had the initial development of invariant models by featurizing only the edge distances (Schütt et al., 2018). DimeNet (Gasteiger et al., 2020) and GemNet (Gasteiger et al., 2021) further increased their expressiveness by including invariant bond direction information. EScAIP combines the self-attention mechanism and the BOO features to build an invariant backbone architecture, while having a prediction head without built-in equivariance (Qu & Krishnapriyan, 2024). Finally, there are also models without built-in equivariance. The Orb models explored this

direction and relied on data augmentation and additional loss terms to recover the equivariance (Neumann et al., 2024; Rhodes et al., 2025).

## 3 BSCT: Bond Smoothness Characterization Test

Evaluating an MLIP requires more than measuring its energy and force prediction errors. For an MLIP to make reliable predictions in simulations, the physical soundness of the potential energy surface (PES) it predicts is equally important. In the quantum chemistry community, the *chemical smoothness* of a PES is defined rigorously as the absence of spurious discontinuities, extrema, and inflection points (Subotnik et al., 2008). This definition differs from the MLIP community's more recent use of "smoothness" to mean bounded PES derivatives (Fu et al., 2025), which typically requires costly microcanonical MD simulations to evaluate.

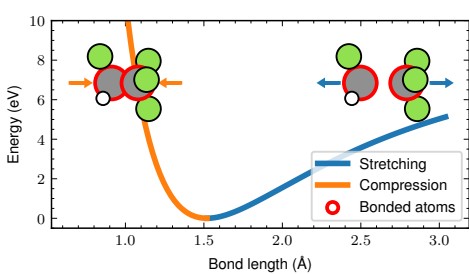

Figure 1: Example of a C-C bond in a $C_2H_2F_4$ molecule from the BSCT-SPICE dataset. The DFT reference PES smoothly varies across the wide range of bond lengths, showing the behavior expected from reliable interatomic potentials.

**Near vs. Far-From-Equilibrium Smoothness.** Existing benchmarks provide valuable measures of smoothness and energy conservation, but they are often focused on near-equilibrium configurations (Fu et al., 2025; Póta et al., 2024). In this regime, a sufficiently expressive MLIP can often improve smoothness by training on more data points. For example, models on the Matbench Discovery leaderboard can do this and achieve better near-equilibrium smoothness ($\kappa_{\mathrm{SRME}}$) with the same architecture (Riebesell et al., 2023). However, far-from-equilibrium smoothness is different. These configurations are true out-of-distributions (OOD) relative to the training set, and adding more near-equilibrium data does not help. They are also rarely explored by MD simulations, making MD-based evaluations infeasible due to computational cost. Meanwhile, datasets such as GMTKN55 include far-from-equilibrium reactions, but they are tailored for *ab initio* methods and do not explicitly assess PES smoothness (Goerigk et al., 2017). This leaves a gap for benchmarks that efficiently evaluate smoothness in far-from-equilibrium regions, and our work addresses this limitation with the Bond Smoothness Characterization Test (BSCT).

**What BSCT Measures.** The Bond Smoothness Characterization Test (BSCT) evaluates how smoothly an MLIP predicts energies and forces as molecular bonds are systematically stretched and compressed beyond equilibrium. We focus on bonds because their ground truth PES (i.e., dissociation curves) is intrinsically smooth, making deviations and erroneous non-smoothness easy to detect. By sampling a one-dimensional slice of the PES along the bond length and comparing MLIP predictions with density functional theory (DFT) references, BSCT isolates non-smooth behavior in regions outside the training distribution. BSCT targets the challenging regime of far-from-equilibrium smoothness, enabling us to determine which specific design choices genuinely improve PES smoothness. This allows us to isolate the effect of architecture from raw model capacity, providing insight into how design choices influence the reliability and generalization of MLIPs.

**Constructing the BSCT Dataset.** For each molecule, we select a bond that splits the molecule into two fragments and displace the fragments along the bond axis while keeping their internal geometries fixed. Formally, given atomic positions $\{x_i\} \in \mathbb{R}^{3 \times N}$, the fragment labels $\{h_i\} \in \{-1, 1\}^N$, and the bond direction unit vector $\hat{r} \in S^2$, the perturbed positions for displacement $\alpha$ are:

$$x'_i(\alpha) = x_i + \alpha h_i \hat{r}. \tag{1}$$

We construct the **BSCT-SPICE dataset** by applying this procedure to the SPICE test structures Eastman et al. (2023); Kovács et al. (2023). The dataset contains 485 molecules, each with 100 DFT single-point calculations computed at the same level of theory as SPICE ($\omega$B97M-D3(BJ)/def2-TZVPPD) using Psi4 (Mardirossian & Head-Gordon, 2016; Rappoport & Furche, 2010; Hellweg & Rappoport, 2015; Turney et al., 2012), the same computational chemistry code as SPICE.

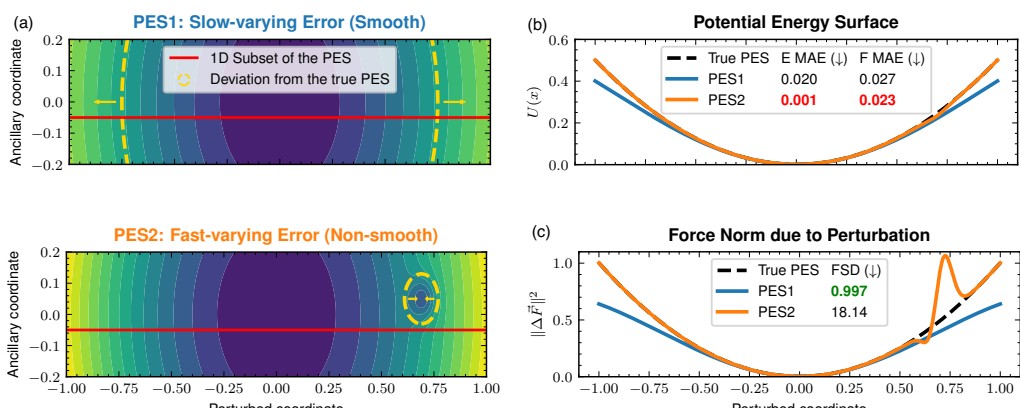

Figure 2: (a): Comparison of two hypothetical PES. Both PES1 and PES2 accurately reproduce the true quadratic PES near equilibrium. Away from equilibrium, PES1 slowly deviates from the quadratic PES reference but remains smooth, and PES2 has an artificial minimum (non-smoothness) enclosed by gold dashed lines. (b) Standard metrics such as energy and forces mean absolute errors (MAEs) evaluated on the one-dimensional subset fail to detect PES2's non-smoothness. (c) Our proposed force smoothness deviation (FSD) metric sensitively captures this non-smooth behavior.

We construct the dataset by systematically scanning bridge bonds in molecules, filtering out structures with isolated or overlapped atoms, and running DFT calculations on bond scans (See Appendix A for details). Figure 1 shows an example C–C bond scan, where the DFT PES varies smoothly across the sampled bond lengths; BSCT evaluates whether MLIPs preserve this physically correct behavior.

**Quantifying Smoothness: The FSD metric.** We introduce a new metric, the Force Smoothness Deviation (FSD), to quantitatively measure the PES smoothness. We define the "force norm due to perturbation" as:

$$\|\Delta \vec{F}\|^2 = \|\vec{F} - \vec{F}_{\min E}\|^2, \tag{2}$$

where $\vec{F}_{\min E}$ is the force vector at the minimum-energy structure in the 1-D PES section. As shown in Figure 2, the derivative of $\|\Delta \vec{F}\|^2$ with respect to the perturbed coordinate $\alpha$ sensitively detects non-smoothness (artificial minimum). Therefore, we define the force smoothness deviation (FSD) as:

$$\text{FSD} = \max_{\alpha} \left| \frac{\frac{d}{d\alpha}\|\Delta \vec{F}_{\text{MLIP}}\|^2}{\|\Delta \vec{F}_{\text{MLIP}}\|^2} - \frac{\frac{d}{d\alpha}\|\Delta \vec{F}_{\text{DFT}}\|^2}{\|\Delta \vec{F}_{\text{DFT}}\|^2} \right| = \max_{\alpha} \left| \frac{d}{d\alpha} \log \frac{\|\Delta \vec{F}_{\text{MLIP}}\|^2}{\|\Delta \vec{F}_{\text{DFT}}\|^2} \right|, \tag{3}$$

where $\alpha$ is defined by Equation 1, and the derivative is taken with respect to this one-dimensional perturbation parameter. A lower FSD indicates smoother and more physically sound PES predictions. The particular functional form makes FSD an indicator of chemical smoothness since the denominator $\|\Delta \vec{F}_{\text{MLIP}}\|^2$ is small when an extremum is around, and the numerator $\frac{d}{d\alpha}\|\Delta \vec{F}_{\text{MLIP}}\|^2$ is small when an inflection point is around. By comparing the ratio of them to the DFT reference, FSD can detect any artificial extrema or inflection points and measure the "smoothness" of a PES.

## 4 DESIGNING AN EXPRESSIVE, UNCONSTRAINED BACKBONE MLIP

To study how specific architectural design choices influence potential energy surface (PES) smoothness, we begin with a flexible backbone that imposes minimal geometric constraints. This neutral starting point ensures that improvements in the FSD metric can be attributed to targeted design choices rather than confounding effects from built-in constraints. While geometric constraints are common in existing architectures, they can limit scalability and expressivity (Sriram et al., 2022), and their connection to PES smoothness remains unclear. Our backbone, the **Min**imally constrained **D**ifferentiable **Sc**aled **A**ttention **I**nteratomic **P**otential, **MinDScAIP**, removes rotational equivariance constraints and incorporates a self-attention mechanism inspired by the Swin-Transformer (Liu et al.,

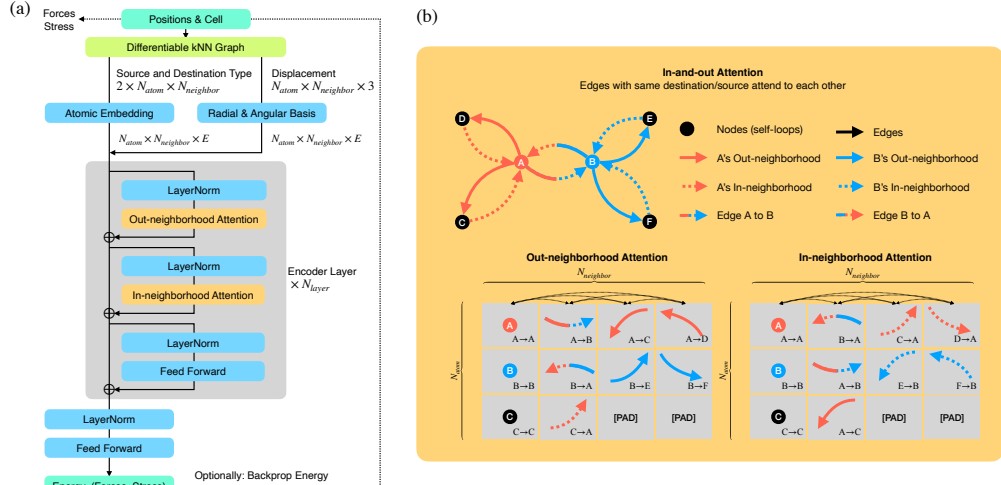

Figure 3: (a): Overview of the MinDScAIP architecture backbone. The top-level architecture is similar to a Transformer encoder with two attention blocks, connected by a pre-norm residual path. (b) Alternating in-and-out neighborhood attention, analogous to the shifted window attention of Swin Transformer. The interleaving windows allow information to propagate across the molecular graph.

2021), originally designed for efficient and scalable vision tasks. Applied to molecular graphs, this design aims to balance high computational efficiency with strong representational capacity.

## 4.1 BACKBONE ARCHITECTURE OVERVIEW

The MinDScAIP backbone, summarized in Figure 3a, differs from conventional MLIP design mainly in the graph construction method and the attention mechanism employed. These designs are proposed to balance expressiveness and computational scalability, while keeping the architecture unconstrained enough to serve as a neutral platform for testing BSCT-guided modifications.

**Graph Construction Methods.** Most attention-based MLIPs impose sparsity by restricting attention to spatially local neighborhoods rather than fully-connected self-attention, often using radius graphs. However, using radius graphs results in a long-tailed distribution of neighbor counts, which can make attention inefficient—either requiring sparse attention kernels or dense attention with substantial padding (see Appendix B). For a more regular and efficient representation, the MinDScAIP backbone adapts a k-nearest-neighbor (kNN) graph. We arrange kNN edges into an $N_{atom} \times k$ array such that $(i, j)$-th element is the $j$-th shortest edge in node $i$'s neighborhood Qu & Krishnapriyan (2024). Essentially, rather than *padding* all neighborhoods to the same number of neighbors, we *truncate* the neighbors to obtain a regular representation. To make the truncation differentiable, we further propose a Differentiable k-NN Algorithm, which will be detailed in Section 4.2.

**Attention Mechanism.** MinDScAIP employs a dual attention strategy inspired by Swin-Transformer's shifted window attention (Liu et al., 2021). Swin-Transformer partitions an image into windows (disjoint local sets). Similarly, on a directed graph, *neighborhoods* can partition edges into disjoint local sets. Similar to Swin-Transformer, where windows are shifted to create overlaps to propagate information, we can switch between in-neighborhood and out-neighborhood to achieve the same goal. As illustrated in Figure 3b, MinDScAIP interleaves attention between edges sharing the same source atom (out-neighborhood) and edges sharing the same destination atom (in-neighborhood), generalizing Swin-Transformer to directed graphs.

## 4.2 DIFFERENTIABLE kNN ALGORITHM

Another critical aspect of physical soundness is a conservative force field. While kNN graphs offer computational advantages, standard kNN algorithms are inherently non-differentiable due to the

truncation. Differentiability is crucial to predicting conservative forces. To address this issue, we introduce the differentiable kNN algorithm (Diff-kNN), which can provide the best of both worlds.

**Standard kNN Algorithm.** The standard kNN algorithm has two steps: first, a ranking is calculated for each edge $(i, j)$ with length $d_{ij}$ with respect to other edges in its out-neighborhood $\mathcal{N}_{out}(i) = \{(i, j')\}_{j'}$. This ranking is *hard*, as rankings can change discontinuously when edge lengths vary.

$$\text{rank}((i, j)|\mathcal{N}_{out}(i)) = \sum_{j'} \mathbb{I}(d_{ij} > d_{ij'}). \tag{4}$$

Second, it selects the $k$ shortest edges:

$$G = \{(i, j) : \text{rank}((i, j)|\mathcal{N}_{out}(i)) < k\} \tag{5}$$

Both the ranking and selection steps are non-differentiable.

**Soft Ranking with Differentiable kNN Algorithm (Diff-kNN).** To preserve differentiability of the kNN graph, we propose the differentiable kNN algorithm (Diff-kNN), which replaces the non-differentiable hard ranking with a differentiable soft ranking using a sigmoid function (see Figure 4):

$$\text{rank}((i, j)|\mathcal{N}_{out}(i)) = \sum_{j'} \sigma((d_{ij} - d_{ij'})/d_0), \tag{6}$$

where $d_0$ is a scale parameter that controls the sharpness of the sigmoid. This soft-ranking algorithm makes the ranking step differentiable [1] Edge selection is made differentiable using a smooth envelope function (Gasteiger et al., 2020; Pozdnyakov & Ceriotti, 2023), which assigns edge weights by:

$$e_{ij} = \exp(-f_{\text{env}}^2/(1 - f_{\text{env}}^2)), \text{where } f_{\text{env}} = \frac{\text{rank}((i, j)|\mathcal{N}_{out}(i))}{k}. \tag{7}$$

$e_{ij}$ smoothly vanishes to zero when $f_{\text{env}} = 1 \Leftrightarrow \text{rank}((i, j)|\mathcal{N}_{out}(i)) = k$. The graph is therefore constructed by selecting all edges with $f_{\text{env}} < 1$. By biasing self-attention by the edge weights, the envelope function can ensure the smoothness of the selection step.

**Combining Diff-kNN with a Radius Cutoff.** To avoid unbounded edge lengths, which can produce long-tailed neighbor distributions, we combine Diff-kNN with a soft radius cutoff $r_c$. To incorporate this feature, we can redefine $f_{\text{env}}$ as:

$$f_{\text{env}} = \beta^{-1} \cdot \log(e^{\beta f_{\text{env, out}}} + e^{\beta f_{\text{env, radius}}}), \tag{8}$$

where $f_{\text{env, out}} = \text{rank}((i, j)|\mathcal{N}_{out}(i))/k$ and $f_{\text{env, radius}} = d_{ij}/r_c$. The log-sum-exponential is a soft approximation of the maximum function, and $\beta$ is a parameter controlling its smoothness.

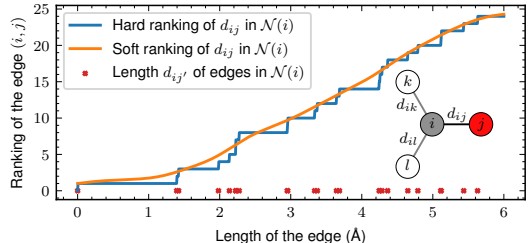

Figure 4: The Diff-kNN algorithm replaces the hard ranking algorithm used in standard kNN with the soft ranking described in Equation 7.

## 5 INVESTIGATING BSCT AS A TOOL FOR MODEL DESIGN

Having established an expressive, unconstrained backbone (Section 4), we now use BSCT to investigate the central question: *Which targeted architectural modifications can improve potential energy surface (PES) smoothness while maintaining scalability and expressiveness?*

Our approach is to start from the neutral backbone MLIP, identify components that may introduce nonlinearities and propose designs to regularize them (Section 5.1), validate the effectiveness of BSCT (Section 5.2), introduce smoothness-oriented design choices independently (Section 5.3 & 5.4), and present the final results (Section 5.5). All hyperparameters can be found in Appendix G.

---

[1]It is memory inefficient when $N_{\text{atom}} \gg 1$. A memory-efficient version is described in Appendix C.

## 5.1 Identification of Design Choices

To establish the relationship between PES smoothness and specific model design choices, we first analyze the backbone architecture and locate components that introduce nonlinearity into the model's predictions. We propose targeted design choices to specifically regularize these nonlinearities, aiming to provide theoretical guarantees of PES smoothness. Each design choice is introduced independently and evaluated on its impact on three aspects of performance: (i) accuracy near equilibrium (energy and force mean absolute errors, MAEs), (ii) smoothness (Force Smoothness Deviation, FSD metric), and (iii) energy conservation in microcanonical molecular dynamics simulations.

**Sources of Nonlinearity.** The MinDScAIP architecture consists of three main parts: featurization, attention blocks (self-attention & feedforward), and prediction heads (feedforward). From these, we identify three main sources of nonlinearities: (1) the Gaussian smearing featurization, (2) nonlinear activation functions, and (3) the softmax in scaled dot-product attention.

**Smoothness-Oriented Design Modifications.** We propose the following regularization strategies:

- **Controllable Gaussian Smearing**: Gaussian smearing featurizes atomic distances using Gaussian kernels (Schütt et al., 2018): $v_i = \exp(-\frac{|d-\mu_i|^2}{2\sigma^2})$, where $\sigma$ is set to be the spacing $\Delta x$ of $\mu_i$. We introduce a scaling factor $\gamma$, setting $\sigma = \gamma \Delta x$. Increasing $\sigma$ upper bounds the derivatives of any linear combination of $v_i$ relative to its infinity norm, thereby improving smoothness (see Appendix D).

- **Weight Decay**: Weight decay is a standard regularization technique that promotes NN smoothness. By regularizing the norm of the NN parameters, inputs to the activation function, such as SiLU, remain small and produce smoother transitions when the input structure is changed.

- **Temperature-controlled Attention**: We introduce a temperature parameter into the scaled dot-product attention mechanism: $\text{Attention}(Q, K, V; \tau) = \text{Softmax}\left(\frac{QK^T}{\tau\sqrt{E_k}}\right)V$. Larger $\tau$ values yield smoother attention outputs. Although temperature can be absorbed by scaling $Q$ and $K$, weight decay limits the magnitude of projection parameters, preventing arbitrary rescaling and making $\tau$ an effective smoothness control.

## 5.2 BSCT as an early indicator of molecular dynamics stability

The Force Smoothness Deviation (FSD) metric from BSCT is new to the community. To assess its physical relevance, we examine whether FSD correlates with stability in far-from-equilibrium molecular dynamics (MD) simulations.

**Problem Setup.** We select molecular structures from the MD22 dataset and relax them to their ground state using the MLIP. The system is equilibrated for 10 ps using a Langevin integrator with friction $1\text{ps}^{-1}$. We then run high-temperature simulations, where bond breaks in far-from-equilibrium geometries. We monitor the kinetic temperature of the system to detect any unrealistic jumps in kinetic energy. Sudden increases in kinetic temperature in a short period ($\gg T_{bath}$ within 10fs) are unlikely to originate from the heat bath and instead suggest spuriously large forces due to PES non-smoothness. We test three MinDScAIP models with the same architecture but varying strengths of smoothness-oriented designs, enabling us to isolate their correlation with FSD values. The study is repeated 10 times with different seeds, resulting in 70 distinct trajectories per model and temperature to increase the statistics.

**Results.** Figure 5 shows that higher FSD values (i.e., the more non-smoothness detected by BSCT) correlate strongly with more frequent and larger kinetic temperature spikes. A more quantitative measure is presented in Table 1, where we calculate the maximum change in kinetic temperature in 10fs and averaged over the 70 trajectories. This supports FSD as an early, low-cost predictive indicator of MD stability in far-from-equilibrium regimes: FSD computation takes $\sim 40$ minutes on one A6000 GPU, while MD simulations take $\sim 40$ hours.

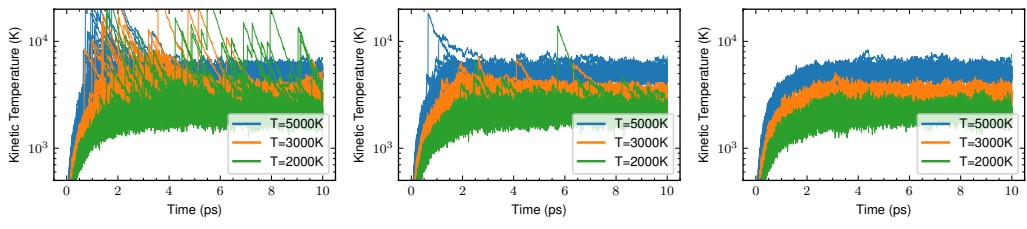

(a) MinDScAIP Vanilla     (b) MinDScAIP w/ Weight Decay    (c) MinDScAIP w/ Smear.&Temp.

Figure 5: The kinetic temperature plotted as a function of time. The seventy trajectories (seven structures from MD22, each with ten random seeds) at the same temperature are shown in the same color. A more quantitative measure can be found in Table 1

Table 1: The maximum change in kinetic temperature in 10fs of each trajectory shown in Figure 5, aggregated by averaging over all trajectories of the same temperature. We see a strong positive correlation between maximum jump and FSD values.

| Ablation | FSD (1/Å) | Max Jump 2000K | Max Jump 3000K | Max Jump 5000K |
|---|---|---|---|---|
| Vanilla | 97.4 | 9734 | 813 | 597 |
| Weight Decay | 76.3 | 1904 | 681 | 509 |
| Smear. & Temp. | 43.2 | 490 | 614 | 514 |

## 5.3 BSCT-SPICE ABLATIONS TO INVESTIGATE PHYSICAL SOUNDNESS

In addition to the above design choices, we investigate how the prediction head and model size impact physical soundness. We conduct controlled ablation studies measuring energy/force MAEs on the SPICE MACE-OFF test split (Eastman et al., 2023; Kovács et al., 2023), energy drift in NVE MD simulations, and the force smoothness deviation (FSD) metric on BSCT-SPICE. Failure to achieve reasonable performance on any of these metrics indicates potential unphysicality in the MLIP. Details of NVE MD energy drift experiments are in Appendix E. The ablations consider:

1. *Prediction head*: direct-force, gradient-based forces (standard kNN), and gradient-based forces (Diff-kNN) with the large model (60M).

2. *Model size*: Gradient-based Diff-kNN models with small (3.8M), medium (15M), and large (60M) parameters.

3. *Smoothness design choices*: Impact of weight decay, Gaussian smearing widths, and temperature-controlled attention.

4. *Baselines to compare against*: MACE Large (4.7M), the baseline model trained on SPICE, and GemNet-T, which recently showed good performance on OMol25 (Levine et al., 2025).

**Results.** Table 2 summarizes the evaluation results. The principal findings are:

1. **Prediction Head**:
   - Direct-force models yield low FSD but violate energy conservation because direct force models with proper normalization never output large forces, resulting in low FSD.
   - Standard kNN shows large NVE energy drift, confirming non-conservative force field..
   - Diff-kNN restore conservative behavior while retaining smoothness.

2. **Model Size**:
   - Larger models improve accuracy but degrade far-from-equilibrium smoothness, consistent with the intuition that additional nonlinearities can cause abrupt PES changes.
   - Regularization is essential for scaling MLIPs.

3. **Smoothness Design Choices**:
   - Increased smearing width smooths compressed-bond regions.
   - Higher attention temperature smooths stretched-bond regions.

- Combining both (Smear. & Temp.) yields the smoothest PES.

4. **Baselines**:

- MinDScAIP with smoothness designs outperforms MACE and GemNet-T in near-equilibrium accuracy while matching their BSCT smoothness scores

Table 2: MinDScAIP performance on SPICE and BSCT. The best model is boldfaced, while the best per group is underlined. For energy drift, since small values for conservative models correspond to numerical errors, the best model is not boldfaced, and non-conservative ones are colored in red.

| Ablation | | Test MAE ($\downarrow$) | | NVE Sim. | FSD ($\downarrow$) | | |
| | | Energy | Forces | Energy Drift | Full | Compress | Stretch |
| Group | Model | (meV/atom) | (meV/Å) | (meV/atom) | (1/Å) | (1/Å) | (1/Å) |
|---|---|---|---|---|---|---|---|
| Prediction Head | Direct Force | 0.15 | 4.22 | 2.6e5 | 71.8 | 69.4 | 42.8 |
| | Gradient kNN | **0.08** | 3.01 | 19.10 | 105 | 80.7 | 93.1 |
| | Gradient Diff-kNN | 0.09 | 3.02 | 0.678 | 97.4 | 69.5 | 87.2 |
| Model Size | Small | 0.23 | 6.63 | 0.640 | 80.2 | 61.3 | 66.2 |
| | Medium | 0.12 | 4.01 | 0.722 | 93.2 | 75.4 | 79.8 |
| | Large | 0.09 | 3.02 | 0.678 | 97.4 | 69.5 | 87.2 |
| Smoothness Design Choices | Weight decay | 0.09 | 2.94 | 0.631 | 76.3 | 65.2 | 55.1 |
| | Smearing | 0.10 | **2.86** | 0.832 | 83.1 | 32.3 | 82.5 |
| | Temperature | 0.10 | 3.06 | 0.675 | 75.5 | 63.3 | 62 |
| | Smear. & Temp. | 0.12 | 2.94 | 0.708 | 43.2 | 32.8 | 38.1 |
| Baseline | MACE | 0.79 | 14.3 | 0.691 | 62.1 | 62.1 | **12.3** |
| | GemNet-T | 0.30 | 7.11 | 110.6 | **33.8** | **28.8** | 20.5 |

## 5.4 NEAR-EQUILIBRIUM SMOOTHNESS: MPTRJ ABLATIONS

We assess whether smoothness-oriented designs benefit near-equilibrium behavior using the MPTrj dataset and the Matbench Discovery benchmark, noting that the smoothness design choices are not specific to far-from-equilibrium systems and are applicable to improve near-equilibrium smoothness.

**Problem Setup.** We train three versions of MinDScAIP-30M on the MPTrj dataset, with weak, moderate, and strong smoothness-oriented designs. The MLIPs trained on MPTrj are evaluated on the Matbench Discovery benchmark (Riebesell et al., 2023), which tests model capability to relax materials to their ground state geometry (RMSD), correctly predict their stability (F1), and capture phonon modes ($\kappa_{\mathrm{SRME}}$) (Póta et al., 2024). Models are pretrained with direct-force and DeNS targets (Liao et al., 2024), then fine-tuned with a gradient-based prediction head following Fu et al. (2025). We focus on the $\kappa_{\mathrm{SRME}}$ metric, which reflects the smoothness near equilibrium.

**Results.** Table 3 shows that stronger smoothness designs yield modest F1 improvements but substantial $\kappa_{\mathrm{SRME}}$ reductions. Since $\kappa_{\mathrm{SRME}}$ requires calculating a dense grid of force sets, FSD again serves as a faster, cheaper proxy to evaluate such smoothness in the process of model development.

## 5.5 FINAL RESULTS

Given the insights from the BSCT and MPTrj ablations, we present our final results on the SPICE and MPTrj datasets in Table 4 and 5, respectively. MinDScAIP with smoothness-oriented design achieves strong performance on both datasets while being physically sound and scalable (see Appendix F for

Table 3: MinDScAIP MPTrj ablation studies results. The three MLIPs with weak, moderate, and strong smoothness designs are listed with their hyperparameters.

| Model | Weight Decay | Temperature | Smearing Width | F1 $\uparrow$ | $\kappa_{\mathrm{SRME}}$ $\downarrow$ | RMSD $\downarrow$ |
|---|---|---|---|---|---|---|
| Weak | $1 \times 10^{-3}$ | 1 | 1 | 0.807 | 0.77 | 0.092 |
| Moderate | $1 \times 10^{-2}$ | 5 | 5 | 0.811 | 0.63 | 0.089 |
| Strong | $5 \times 10^{-2}$ | 10 | 10 | 0.817 | 0.49 | 0.088 |

Table 4: MinDScAIP Smear. & Temp. performance on the SPICE test set, binned by the datasets of molecules. The errors are reported in per-atom energy MAE (meV/atom) and forces MAE (meV/Å).

| Dataset | MACE 4.7M | | EScAIP 45M | | eSEN 6.5M | | MinDScAIP 60M | |
|---|---|---|---|---|---|---|---|---|
| | **E** | **F** | **E** | **F** | **E** | **F** | **E** | **F** |
| PubChem | 0.88 | 14.75 | 0.53 | 5.86 | 0.15 | 4.21 | **0.14** | **3.38** |
| DES370K M. | 0.59 | 6.58 | 0.41 | 3.48 | 0.13 | 1.24 | **0.06** | **0.99** |
| DES370K D. | 0.54 | 6.62 | 0.38 | 2.18 | 0.15 | 2.12 | **0.09** | **0.90** |
| Dipeptides | 0.42 | 10.19 | 0.31 | 5.12 | 0.25 | 3.68 | **0.09** | **1.48** |
| Solvated A.A. | 0.98 | 19.43 | 0.61 | 11.52 | 0.25 | **3.68** | **0.13** | 3.96 |
| Water | 0.83 | 13.57 | 0.72 | 10.31 | 0.15 | 2.50 | **0.13** | **2.29** |
| QMugs | 0.45 | 16.93 | 0.41 | 8.74 | **0.12** | 3.78 | 0.16 | **2.86** |

Table 5: Evaluation on the Matbench Discovery benchmark. Models are sorted in F1 order, with our model (MinDScAIP-60M) at the top. The row below it shows the phonon calculation with 0.1Å finite displacement instead of the default of 0.03Å. Increasing the finite displacement spacing can suppress the local fluctuations and improve phonon prediction quality (Fu et al., 2025).

| Model | F1 ↑ | DAF ↑ | Precision ↑ | Accuracy ↑ | MAE ↓ | R2 ↑ | $\kappa_{SRME}$ ↓ | RMSD ↓ | CPS ↑ |
|---|---|---|---|---|---|---|---|---|---|
| MinDScAIP-60M | 0.833 | 5.313 | 0.812 | 0.948 | 0.035 | 0.789 | 0.691 | 0.0845 | 0.722 |
| (0.1Å-displacement) | 0.833 | 5.313 | 0.812 | 0.948 | 0.035 | 0.789 | 0.340 | 0.0845 | 0.792 |
| eSEN-30M-MP | 0.831 | 5.260 | 0.804 | 0.946 | 0.033 | 0.822 | 0.340 | 0.0752 | 0.797 |
| eqV2 S DeNS | 0.815 | 5.042 | 0.771 | 0.941 | 0.036 | 0.788 | 1.676 | 0.0757 | 0.522 |
| MatRIS-MP | 0.809 | 5.049 | 0.772 | 0.938 | 0.037 | 0.803 | 0.861 | 0.0773 | 0.681 |
| DPA3-v2-MP | 0.786 | 4.822 | 0.737 | 0.929 | 0.039 | 0.804 | 0.959 | 0.0823 | 0.718 |
| Eqnorm MPtrj | 0.756 | 4.844 | 0.741 | 0.929 | 0.040 | 0.799 | 0.408 | 0.084 | 0.756 |
| ORB v2 MPtrj | 0.765 | 4.702 | 0.719 | 0.922 | 0.045 | 0.756 | 1.725 | 0.1007 | 0.470 |
| Nequip-MP-L | 0.761 | 4.704 | 0.719 | 0.921 | 0.043 | 0.791 | 0.452 | 0.086 | 0.733 |
| SevenNet-l3i5 | 0.760 | 4.629 | 0.708 | 0.920 | 0.044 | 0.776 | 0.550 | 0.0847 | 0.714 |
| GRACE-2L-MPtrj | 0.691 | 4.163 | 0.636 | 0.896 | 0.052 | 0.741 | 0.525 | 0.0897 | 0.681 |
| MACE-MP-0 | 0.669 | 3.777 | 0.577 | 0.878 | 0.057 | 0.697 | 0.647 | 0.0915 | 0.644 |

inference efficiency benchmark). This demonstrates the utility of BSCT as a practical, computationally efficient tool for guiding MLIP development.

## 6 CONCLUSION

We introduce the Bond Smoothness Characterization Test (BSCT) as a targeted, low-cost diagnostic of potential energy surface (PES) smoothness predicted by machine-learned interatomic potentials (MLIPs), enabling early detection of near- and far-from-equilibrium instabilities, as evidenced by correlations with MD simulations and thermal conductivity calculations. Through a principled investigation guided by BSCT, we identified architectural modifications—such as differentiable kNN graphs, adjustable smearing width, and temperature-controlled attention—that improve PES smoothness while preserving scalability and expressiveness. More broadly, BSCT demonstrates how physics-motivated evaluation metrics can directly inform model design, providing a practical framework for developing MLIPs that combine accuracy, scalability, and physical soundness.

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

## A    BSCT SAMPLING PROCEDURE

The BSCT-SPICE dataset is constructed with the following procedure.

1. Identify all candidate bridge bonds that partition the molecule into two separate fragments without creating isolated atoms.

2. Linearly stretch and compress candidate bonds, covering bond lengths from $0.5\times$ to $2\times$ the sum of the bonded atoms' covalent radii.

3. Exclude candidate bonds if, upon perturbation, any pair of atoms (except the pair bonded by the candidate) becomes closer than $0.9\times$ the sum of the bonded atoms' covalent radii.

4. Sample from the filtered dataset across selected bond types (C–C, C–N, C–O, C–P, C–S, N–N, N–O, N–P, and O–P).

5. Run DFT on 100 evenly spaced structures along each selected bond perturbation trajectory.

6. Exclude bonds with discontinuous PES due to Self-Consistent Field convergence issues.

Since the sampling process is random, we counteracted the arbitrariness by controlling the bond types that we sample, such as C-C, C-O, etc., since bond types are the most indicative property of a bond. We also set the bond lengths of the bridge bond independently by the sum of the covalent radii of the bonded atoms instead of using the bond length in the original structure.

## B    GRAPH CONSTRUCTION METHODS

Graph construction methods have a critical impact on the topology of the graphs. In most settings, MLIPs use a radius graph to impose locality constraints and implement periodic boundary conditions. However, the distribution of the number of neighbors is usually very long-tailed. In Figure 6a, the number of neighbors as a function of radius cutoff is calculated for the SPICE MACE-OFF dataset. The maximum number of neighbors increases significantly faster than the average case. As discussed in the main text, if we want to use dense attention kernel, we must pad the sequences to the same length, the maximum number of neighbors. Figure 6b shows an illustration of an array with a large padding rate. The majority of the computational power is spent on the padding tokens, which do not influence the results. Using k-Nearest-Neighbors to limit the maximum number of neighbors can close the gap between the mean and the maximum number of neighbors, which is highly preferable when using dense attention kernels.

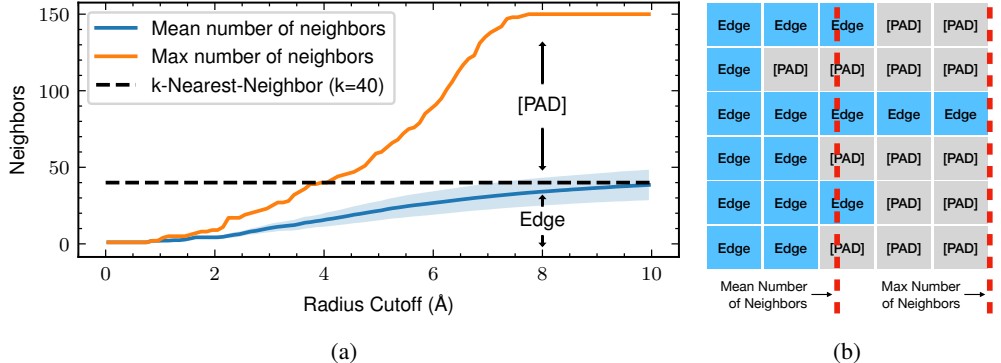

(a)                                                                                    (b)

Figure 6: (a): Mean (± standard deviation) and maximum number of neighbors as a function of radius cutoff in a radius graph. To use a dense implementation of attention, one must pad all sequences to the same length, which is the maximum number of neighbors. (b) Illustration of padding inefficiency due to the long-tailed distribution of neighbor counts. Since the padding also requires computation, ideally, we want to minimize the gap between the two red dashed lines.

## C  MEMORY EFFICIENT DIFF-kNN

To make the Diff-kNN algorithm discussed in Section 4.2 memory efficient, we observe that the soft ranking function,

$$\text{rank}((i,j)|\mathcal{N}_{out}(i)) = \sum_{j'} \sigma((d_{ij} - d_{ij'})/d_0), \tag{9}$$

where $\sigma(x)$ is the sigmoid function, $d_{ij}$ are the lengths of the edges, and $d_0$ controls the sharpness, requires $O(N_{\text{atoms}}^2)$ space for each calculation since each term $\sigma((d_{ij} - d_{ij'})/d_0)$ contribute non-zero values to the sum. To save space, we would like to truncate the summation. To achieve this, we can replace $\sigma(x) = \frac{1}{1+e^{-x}}$ with the bump function $g(x)$, whose derivative has compact support on $[-1,1]$:

$$g(x) = \begin{cases} 0 & \text{if } x < -1 \\ \frac{e^{-2/(x+1)}}{e^{-2/(x+1)}+e^{-2/(x-1)}} & \text{if } x \in [-1,1] \\ 1 & \text{if } x > 1 \end{cases} \tag{10}$$

where $g(x) \approx x$ when $|x| \ll 1$ and $g(x) \equiv \mathbb{I}(x)$ when $|x| > 1$. Since $g(x)$ is strictly zero when $x < -1$, we can sort edge lengths and truncate at some rank $k + \Delta$ before taking the summation, since the edges with large displacement will contribute 0 to the sum when calculating the envelope factor for the edges with small displacement that will be included in the graph. This can reduce the memory complexity of the gradient graph from $O(N_{\text{atoms}}^3)$ to $O(N_{\text{atoms}}(k + \Delta)^2 + N_{\text{atoms}}^2)$.

## D  GAUSSIAN SMEARING

In this section, we look at Gaussian smearing, as described in Section 5.1. We show that increasing $\sigma$ upper bounds the derivatives of any linear combination of these basis functions relative to their infinity norm. We consider the derivative of an arbitrary linear combination of basis functions $v_i(x) = e^{-(x-i)^2/2\sigma^2}$ specified by coefficients $\{a_i\}$: $f(x) = \sum_i a_i v_i(x)$, where the basis function has width $\sigma$. Now we are interested in the maximum derivative at some location $y$ normalized by the infinity norm of such a linear combination. Without loss of generality, we can assume $y = 0$.

$$\max_{\{a_i\}} \frac{\frac{\partial}{\partial y} \sum_{i=-\infty}^{\infty} a_i e^{-(y-i)^2/2\sigma^2}\big|_{y=0}}{\left\|\sum_{i=-\infty}^{\infty} a_i e^{-(x-i)^2/2\sigma^2}\right\|_\infty} = \max_{\{a_i\}} \frac{\frac{1}{\sigma^2} \sum_{i=-\infty}^{\infty} i a_i e^{-i^2/2\sigma^2}}{\left\|\sum_{i=-\infty}^{\infty} a_i e^{-(x-i)^2/2\sigma^2}\right\|_\infty} \tag{11}$$

$$\approx \frac{1}{\sigma^2} \max_{a(\cdot)} \frac{\int_{-\infty}^{\infty} a(z) x e^{-z^2/2\sigma^2} \mathrm{d}z}{\left\|\int_{-\infty}^{\infty} a(z) e^{-(x-z)^2/2\sigma^2} \mathrm{d}z\right\|_\infty} \tag{12}$$

$$= \frac{1}{\sigma} \max_{a(\cdot)} \frac{\int_{-\infty}^{\infty} a(\sigma z) z e^{-z^2/2} \mathrm{d}z}{\left\|\int_{-\infty}^{\infty} a(\sigma z) e^{-(x-z)^2/2} \mathrm{d}z\right\|_\infty} \tag{13}$$

$$\propto \frac{1}{\sigma} \tag{14}$$

For the first equality, the numerator is derived and evaluated at $x = 0$. After that, the summation is approximated by an integral, and the coefficients become a continuous function. Subsequently, we change the variable of integration. The RHS has no dependence on $\sigma$ other than the $\sigma^{-1}$ factor since $a(\sigma z)$ dependence can be absorbed by the maximization over $a(\cdot)$. We conclude that this quantity is upper bounded by $O(\sigma^{-1})$. Therefore, the larger the smearing width, the more bounded the derivatives are.

## E  ENERGY CONSERVATION TESTS IN MOLECULAR DYNAMICS (MD)

**Problem Setup.** We assess the differentiability and smoothness of our models by performing MD simulations under the microcanonical (NVE) ensemble with the Verlocity Verlet integrator (Larsen et al., 2017). The degree of non-conservative behavior is upper bounded by the magnitude of the higher derivatives of the PES (Hairer et al., 2003). Thus, even if a model is infinitely differentiable, it can still manifest non-conservative behavior if the higher derivatives are not bounded. The "smoothness"

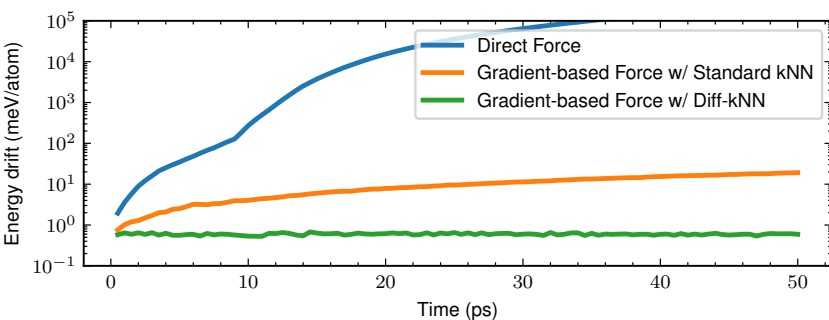

Figure 7: The energy drift of three different MinDScAIP models during microcanonical (NVE) ensembles averaged over seven MD22 molecular trajectories. The direct-force and non-differentiable kNN gradient models (blue and orange curves) show significant energy drift due to non-conservative or discontinuous predictions. The gradient-based Diff-kNN model employs the Diff-kNN algorithm described in Section 4.2 and conserves the energy (the non-zero drift is due to the first-step error of the Verlet integrator).

here is defined as the boundedness of PES derivatives, which is different from the force smoothness deviation defined by BSCT. We follow the protocol proposed by Fu et al. (2025) and use the seven molecules in the MD22 dataset (Chmiela et al., 2023) as out-of-distribution test systems for the MD simulation data relative to SPICE training. Each simulation integrates dynamics for 100ps with 1fs time steps.

**Results.** Figure 7 compares energy drift among MinDScAIP models with different prediction heads and graph structures. The direct force MinDScAIP model cannot conserve energy due to its inherently non-conservative nature. The gradient-based force MinDScAIP model with a standard non-differentiable kNN graph cannot conserve energy due to the piecewise continuity of the standard kNN graph, which results in unbounded first and higher-order derivatives. In contrast, the gradient-based force MinDScAIP model with a differentiable kNN graph can conserve energy. It is infinitely differentiable, and this also demonstrates that its higher derivatives are bounded.

## F INFERENCE EFFICIENCY BENCHMARKS

**Benchmark Setup.** We follow the benchmark described in Fu et al. (2025) to test the throughput and memory usage of MinDScAIP, MACE-MP-16M, and eSEN-30M-OAM (smaller eSEN model weights are not public) on the diamond system. We vary the number of supercells included in the image to test model's throughput and memory usage as a function of number of atoms. All benchmark is done on a single 80GB A100.

**Results** Table 6 summarizes the inference benchmark results. We highlight that MinDScAIP-60M is substantially outperforms MACE-MP-16M in material stability predictions (Matbench F1) by including more parameters (4x the size of MACE-MP-16M), while being just slightly slower than MACE-MP-16M. Comparing against eSEN-30M-OAM, MinDScAIP offers comparable accuracy while being order-of-magnitude faster and memory-efficient than eSEN. Both results suggests that MinDScAIP is more scalable than the current architectures.

Table 6: Inference efficiency benchmark of three selected models. Models are tested on the diamond system with varying number of supercells. The benchmark is done on a single 80GB A100. MinDScAIP promises accuracy similar to eSEN while being just slightly slower than MACE-MP.

| Number of Atoms | Throughput (Millions of Steps Per Day) | | | Memory Usage (GB) | | |
|---|---|---|---|---|---|---|
| | MinDScAIP-60M | MACE-MP-16M | eSEN-30M-OAM | MinDScAIP-60M | MACE-MP-16M | eSEN-30M-OAM |
| 216 | 1.15 | 1.49 | 0.09 | 4.32 | 2.3 | 35.81 |
| 512 | 0.51 | 0.72 | OOM | 9.83 | 5.19 | OOM |
| 1000 | 0.26 | 0.37 | OOM | 19.7 | 10.0 | OOM |
| 1728 | 0.15 | 0.23 | OOM | 35.35 | 17.16 | OOM |
| 2744 | 0.09 | 0.15 | OOM | 59.57 | 27.06 | OOM |

## G  HYPERPARAMETERS

The hyperparameters used for the experiments are summarized in Table 7. Since we adopt the training procedure proposed in Fu et al. (2025), the MPTrj runs involve a pretrain-finetune workflow. The parameters for pretraining are specified by the parentheses.

Table 7: The hyperparameters used for MinDScAIP experiments. MPTrj experiments follow the same training procedure as proposed in (Fu et al., 2025), where a direct force and DeNS (Liao et al., 2024) pretraining takes place, and a conservative fine-tuning follows. The parameters used for pretraining are indicated by parentheses: direct force prediction is enabled during pretraining (✓ in the parentheses) and gradient-based prediction is used for fine-tuning (✗ that follows the parentheses).

| dataset | SPICE MACE-OFF | | | | | | | | | | MPTrj-ablations | | | |
|---|---|---|---|---|---|---|---|---|---|---|---|---|---|---|
| hyperparameter | Direct-Force | Grad. kNN | Grad. Diff-kNN | Small | Medium | Large | W. Decay | Smearing | Temperature | Smear. & Temp. | Weak | Medium | Strong | final-60M |
| Embedding dimension | 512 | 512 | 512 | 128 | 256 | 512 | 512 | 512 | 512 | 512 | 256 | 256 | 256 | 512 |
| Hidden factor | 2 | 2 | 2 | 2 | 2 | 2 | 2 | 2 | 2 | 2 | 2 | 2 | 2 | 2 |
| Number of layers | 8 | 8 | 8 | 8 | 8 | 8 | 8 | 8 | 8 | 8 | 8 | 8 | 8 | 8 |
| Number of heads | 32 | 32 | 32 | 8 | 16 | 32 | 32 | 32 | 32 | 32 | 16 | 16 | 16 | 32 |
| kNN k | 30 | 30 | 30 | 30 | 30 | 30 | 30 | 30 | 30 | 30 | 30 | 30 | 30 | 30 |
| Diff-kNN | ✗ | ✗ | ✓ | ✓ | ✓ | ✓ | ✓ | ✓ | ✓ | ✓ | ✓ | ✓ | ✓ | ✓ |
| Diff-kNN $d_0$ | N/A | N/A | 0.2Å | 0.2Å | 0.2Å | 0.2Å | 0.2Å | 0.2Å | 0.2Å | 0.2Å | 0.2Å | 0.2Å | 0.2Å | 0.2Å |
| Diff-kNN $\beta$ | N/A | N/A | 10 | 10 | 10 | 10 | 10 | 10 | 10 | 10 | 10 | 10 | 10 | 10 |
| Mem-eff. Diff-kNN $\Delta$ | N/A | N/A | N/A | N/A | N/A | N/A | N/A | N/A | N/A | N/A | 20 | 20 | 20 | 20 |
| Radius cutoff | 6Å | 6Å | 6Å | 6Å | 6Å | 6Å | 6Å | 6Å | 6Å | 6Å | 6Å | 6Å | 6Å | 6Å |
| Number of radial basis | 128 | 128 | 128 | 128 | 128 | 128 | 128 | 128 | 128 | 128 | 128 | 128 | 128 | 128 |
| Smearing Scale | 1 | 1 | 1 | 1 | 1 | 1 | 1 | 5 | 1 | 5 | 1 | 5 | 10 | 10 |
| Angular $l_{max}$ | 5 | 5 | 5 | 5 | 5 | 5 | 5 | 5 | 5 | 5 | 5 | 5 | 5 | 5 |
| Temperature $\tau$ | 1 | 1 | 1 | 1 | 1 | 1 | 1 | 1 | 10 | 10 | 1 | 5 | 10 | 10 |
| Direct-Force Prediction | ✓ | ✗ | ✗ | ✗ | ✗ | ✗ | ✗ | ✗ | ✗ | ✗ | (✓)✗ | (✓)✗ | (✓)✗ | (✓)✗ |
| Batch size | 128 | 128 | 128 | 128 | 128 | 128 | 128 | 128 | 128 | 128 | 256 | 256 | 256 | (128)256 |
| Weight decay | $10^{-3}$ | $10^{-3}$ | $10^{-3}$ | $10^{-3}$ | $10^{-3}$ | $10^{-3}$ | $5 \times 10^{-2}$ | $5 \times 10^{-2}$ | $5 \times 10^{-2}$ | $5 \times 10^{-2}$ | $10^{-3}$ | $10^{-2}$ | $5 \times 10^{-2}$ | $10^{-2}$ |
| Warmup factor | 0.2 | 0.2 | 0.2 | 0.2 | 0.2 | 0.2 | 0.2 | 0.2 | 0.2 | 0.2 | (0.2)0. | (0.2)0. | (0.2)0. | (0.2)0. |
| Warmup epochs | 1 | 1 | 1 | 1 | 1 | 1 | 1 | 1 | 1 | 1 | 1 | 1 | 1 | 1 |
| Number of Epochs | 100 | 100 | 100 | 100 | 100 | 100 | 100 | 100 | 100 | 100 | (60)40 | (60)40 | (60)40 | (60)40 |
| EMA decay | 0.999 | 0.999 | 0.999 | 0.999 | 0.999 | 0.999 | 0.999 | 0.999 | 0.999 | 0.999 | 0.999 | 0.999 | 0.999 | 0.999 |
| Gradient Clipping | 100 | 100 | 100 | 100 | 100 | 100 | 100 | 100 | 100 | 100 | 100 | 100 | 100 | 100 |
| Energy Loss Weight | 1 | 1 | 1 | 1 | 1 | 1 | 1 | 1 | 1 | 1 | 1 | 1 | 1 | 1 |
| Forces Loss Weight | 2 | 2 | 2 | 2 | 2 | 2 | 2 | 2 | 2 | 2 | (2)4 | (2)4 | (2)4 | (2)4 |
| Stress Loss Weight | N/A | N/A | N/A | N/A | N/A | N/A | N/A | N/A | N/A | N/A | 10 | 10 | 10 | 10 |

## H  ETHICS STATEMENT

The authors uphold the code of ethics and are committed to maintaining trustworthiness and transparency in their scientific practices. Although this work is expected to have limited direct social impact, we recognize the possibility of its misuse. We therefore urge readers to exercise careful judgment when applying, deploying, or releasing any products or artifacts derived from our work.

## I  REPRODUCIBILITY STATEMENT

The authors are committed to ensuring the reproducibility of this work. All essential technical details are provided in the main text and appendix. In addition, the authors plan to release the code, configurations, datasets, and selected checkpoints upon publication to facilitate independent verification and further research.

