# OpenReview forum: "From Evaluation to Design: Using Potential Energy Surface Smoothness Metrics to Guide MLIP Architectures"
_ICLR.cc/2026/Conference — Submitted to ICLR 2026_

### Official Review · Reviewer_EEZ4 · 2025-10-15

**Soundness:** 3
**Presentation:** 3
**Contribution:** 3
**Rating:** 6
**Confidence:** 5

**Summary:**

This paper introduces the Bond Smoothness Characterization Test (BSCT) — a new benchmark and metric designed to directly measure the smoothness of the potential energy surface (PES) predicted by machine-learned interatomic potentials (MLIPs).
The authors define a quantitative indicator called Force Smoothness Deviation (FSD), which evaluates how smoothly forces vary under systematic bond stretching/compression, providing a cheap proxy for PES regularity both near and far from equilibrium.

**Strengths:**

BSCT/FSD provides a direct, quantitative, and low-cost way to assess the physical smoothness of potential energy surfaces (PES), addressing the long-standing gap between accuracy-based and physics-based MLIP evaluation.

The correlation analysis between FSD and kinetic temperature spikes is elegant and convincingly supports FSD as an early physical reliability predictor.

**Weaknesses:**

It remains unclear whether BSCT/FSD behaves consistently for modern equivariant architectures(e.g., eSCN, ViSNet)

The “temperature-controlled attention” mechanism is intuitively justified but lacks formal analysis of how τ influences Lipschitz continuity or gradient smoothness.

BSCT currently probes only bond-length variations. It does not account for angular or torsional degrees of freedom, which may dominate PES roughness in larger or periodic systems.

**Questions:**

Can FSD be generalized to multi-dimensional collective coordinates (angles, torsions) rather than just single bonds?

How sensitive is FSD to the bond-type distribution or sampling density in the BSCT dataset?

Does incorporating FSD as a differentiable regularizer during training further improve PES smoothness, or is it used purely post-hoc?

What are the computational overheads of Diff-kNN compared with standard radius graphs at scale (e.g., MPTrj supercells)?

---

> ### Author Response · Authors · 2025-11-24
>
> > It remains unclear whether BSCT/FSD behaves consistently for modern equivariant architectures(e.g., eSCN, ViSNet)
>
> We are not sure what the reviewer meant by “consistently.” The construction of BSCT and FSD does not depend on details of the MLIP under test, and thus should be applicable to equivariant architectures as well. We have also tested two equivariant and invariant architectures, namely MACE and GemNet, and reported their performance in the paper.
>
> > The “temperature-controlled attention” mechanism is intuitively justified but lacks formal analysis of how τ influences Lipschitz continuity or gradient smoothness.
>
> It can be shown that the gradient descent weight decay is equivalent to a Gaussian prior of variance inversely proportional to the strength of weight decay. The variance of the attention logits, defined as some unit variance input vector, projected by Key and Query matrices, and taking inner product, will be inversely proportional strength of weight decay squared under the prior. Thus, by scaling the attention logits with a temperature τ, we can control the width of the prior over attention logits, which can be translated into other measures of smoothness.
>
> > BSCT currently probes only bond-length variations. It does not account for angular or torsional degrees of freedom, which may dominate PES roughness in larger or periodic systems.
> > Can FSD be generalized to multi-dimensional collective coordinates (angles, torsions) rather than just single bonds?
>
> In principle, one can construct FSD to an arbitrary dimension. One must define the perturbation parameters, construct the grid, compute force norms due to perturbation for these grid points, and finally take the gradient norm with respect to the perturbation coordinates. However, such generalizations may not be practical, since they would require a significant amount of computation. On the other hand, using FSD for angle and torsion scans is also not as relevant. We propose the FSD metric since we are testing far-from-equilibrium configurations, where MLIPs cannot quantitatively predict the right dynamics, and we focus on qualitative measures of physical soundness. However, as angle and torsion scans are near-equilibrium, a more direct comparison to ground-truth labeling is possible, and this has been done in existing work like TorsionNet-500 [1].
>
> > How sensitive is FSD to the bond-type distribution or sampling density in the BSCT dataset?
> Does incorporating FSD as a differentiable regularizer during training further improve PES smoothness, or is it used purely post-hoc?
>
> We calculated the standard deviation of the FSD evaluation results for each model. The standard deviation is around $20Å^{-1}$. The BSCT dataset we presented includes 485 systems and 48,500 datapoints. Therefore, the statistical uncertainty associated with the final FSD number is about $\frac{20}{\sqrt{485 - 1}}\sim 0.9Å^{-1}$. Given that the gap between different models are at the order of $10Å^{-1}$, we consider that the statistics of the dataset is large enough to suppress the statistical error associated with the random construction process. Meanwhile, FSD numbers also depend on the bond types, but this dependency is also model-specific: some models struggle with certain bond types, while others excel with these bond types. Therefore, we balanced the sampling of the bond types such that such dependencies will be integrated out in a controlled manner.
>
> We did not use FSD as a differentiable regularizer during training. Firstly, evaluation of FSD requires a specific input configuration along with ground truth labelling, which requires changing the training distribution. Since we want an apple-to-apple comparison with other models trained on SPICE, we cannot do this. Secondly, the goal of BSCT is to identify transferable design choices. Optimizing as regularization would indeed give better FSD, but would not help in terms of identifying design choices.
>
>
> [1] Rai, Brajesh K., et al. "Torsionnet: A deep neural network to rapidly predict small-molecule torsional energy profiles with the accuracy of quantum mechanics." Journal of Chemical Information and Modeling 62.4 (2022): 785-800.

---

> > ### Comment · Reviewer_EEZ4 · 2025-11-24
> >
> > The authors mention that “BSCT/FSD does not depend on model details and should apply to equivariant architectures as well,” and present results on MACE and GemNet. While this is helpful, my concern is slightly different.
> >
> > It remains unclear whether the proposed BSCT/FSD protocols behave consistently for more recent, state-of-the-art equivariant architectures such as eSCN, ViSNet, or other high-performing models that have become dominant in current MLIP research.
> >
> > The models evaluated in the manuscript—although representative—are somewhat older generations. Given the rapid progress in equivariant neural networks, it would strengthen the paper if the authors could clarify (or ideally demonstrate) whether BSCT/FSD still reveal similar robustness patterns for these newer architectures.

---

> > > ### Author Response · Authors · 2025-11-25
> > >
> > > We thank the reviewer for the comment. We note that eSCN was proposed in 2023 and is no longer state-of-the-art. Its successor, eSEN, is too expensive to train and hyperparameter-tune from scratch, and there was no public checkpoint provided for the eSEN model trained on SPICE. Similarly, the VisNet model does not have checkpoints or hyperparameters for the SPICE dataset, and so the computational cost for us to train and hyperparameter-tune the model from scratch is high.
> > >
> > > Our reasons for choosing GemNet-T and MACE are as follows: the GemNet-T model has gone through a number of improvements over the last year, and the latest version is one of the most competitive models on the OMol25 leaderboard. Similarly, the MACE model has gone through some updates as well and the checkpoints for the MACE model trained on SPICE are easily available. Both of these models are still competitive across many MLIP benchmarks, and so we believe that they are good representatives of the current status of invariant and equivariant MLIPs.

---

> ### Author Response · Authors · 2025-11-24
>
> > What are the computational overheads of Diff-kNN compared with standard radius graphs at scale (e.g., MPTrj supercells)?
>
> In computational complexity terms, standard radius graphs would have $O(N_i N_a^2) $ complexity, where $N_i$ is the number of images included in the radius cutoff, and $N_a$ is the number of atoms in the image. On the other hand, Diff-kNN has computational complexity of $O(N_i N_a^2 + N_a k^2)$ if using the memory efficient implementation, where $k$ is the number of neighbors.
>
> In practice, the Diff-kNN algorithm is also comparable to standard radius graphs. Benchmarking on a random batch of 100 MPTrj systems, with radius=5 and max neighbor of 30, Diff-kNN takes 132.793ms, fairchem’s radius_graph_pbc_v2 takes 127.922ms, and fairchem’s radius_graph_pbc takes 35.957ms. Note that radius_graph_pbc is very memory-inefficient and could cause OOM for large cells, while Diff-kNN and radius_graph_pbc_v2 do not have this issue.

---

### Official Review · Reviewer_A4yb · 2025-10-27

**Soundness:** 2
**Presentation:** 3
**Contribution:** 3
**Rating:** 6
**Confidence:** 4

**Summary:**

The authors propose the Bond Smoothness Characterization Test (BSCT) to assess smoothness in potential energy surfaces predicted by machine-learned interatomic potentials (MLIPs). This test involves a new metric (the FSD -- force smoothness deviation), which is able to characterize smoothness and correlates well with other tests in most cases. Using this metric, as well as other tests, the authors tune their proposed architecture (MinDScAIP) to achieve very good performance on the SPICE and MPTrj datasets.

**Strengths:**

The BSCT test is novel and original, and the same goes for the MinDScAIP architecture. The paper proposes some interesting ideas and shows results that will be valuable for the community. The quality of the investigation is high and the presentation is clear. The empirical performance of the model is strong.

**Weaknesses:**

The paper and its content are of good quality. However, I believe they are lacking in some areas at the moment.

**Presentation**
- [Minor] The paper proposes two distinct novelties: the BSCT test on one side, and the MinDScAIP architecture on the other. While the sections are well-organized, the abstract does not mention the name of the new architecture, the introduction does not either, and the glue text between the different sections is often vague. For example, it would be helpful, in some points, to clearly state something along the lines of "our contribution is two-fold:...", and/or "After having established the BSCT test as a reliable method for the estimation of PES smoothness, we now proceed to...".
- [Minor] In Figure 5, it seems that the simulations have not equilibrated yet by the time the MD trajectory begins (it takes about another 20 ps for the simulation to reach the target temperature).
- [Minor] Typos: citations in lines 147-148 should be enclosed in paretheses, "adapts" -> "adopts" (line 215), double period in line 392, "bond breaks" is likely a typo in line 347, Table 5 probably contains numbers in *million* time steps per day.
- [Minor] The authors should definitely acknowledge Pozdnyakov 2023 (NeurIPS), who proposes a similar attention-based and unconstrained architecture and whose mechanism for reference frame smoothness resembles the Diff-kNN the authors propose here.

**Rigor**
- At many points the authors claim that MD is expensive (e.g., line 53: "requires costly MD to evaluate", lines 112-113: "which typically requires costly microcanonical MD to evaluate"), then they provide a test and a metric that requires first-principles evaluations. These claims should definitely be softened. See also "methods" and "questions".
- In lines 134-135, the authors claim "we focus on bonds because their ground truth PES is intrinsically smooth". This is incorrect (according to the current widespread definition of PES smoothness as "infinite-order differentiability"), as changes in the electronic ground-state along the dissociation curve can make the PES non-differentiable, just like any other cut or section of any ground-state PES. The authors should clarify what they meant here.
- Line 392: "confirming non-conservative force field". The forces are conservative in this case since they are calculated by backpropagation. It is the discontinuities in the PES caused by the standard kNN search, **combined with the finite step sizes of MD**, which cause the energy drift here. In other words, a hypothetical (and impossible in practice) MD performed with infinitely small steps would conserve energy in this case.

**Methods**
- After claiming that MD is slow, the authors propose the BSCT test, which requires first-principle evaluations. Unfortunately, I do not believe the BSCT test is very practical in its current state. If the structures the authors used are made public, it can be turned into a molecular benchmark for smoothness. However, such a benchmark would not be transferable to models created for purposes other than organic molecules, while MD can be performed on arbitrary systems. Most importantly, even if published, the new benchmark would not easily transfer to other levels of quantum chemical theory. Hence, at the moment, I do not see how it could become widely adopted by model developers without asking them to perform new (and expensive) DFT evaluations on their data (even then, that's assuming they work with molecules; no alternative is provided for materials).

**Questions:**

I thank the authors for this important contibution to the field. I have three questions for them:
- How would an architecture developer or a model creator use their BSCT in practice if their DFT settings are not consistent with those used in this work and/or if they are interested in applications other than molecules (materials, catalysis, etc.)?
- On lines 390-391, the authors mention that "models with proper normalization never output large forces". Is this not worrying, considering that forces can be unbounded in practice? The same concern goes for the energy head of these models.
- The results in Table 4 seem to indicate a trade-off between smoothness, which is beneficial especially when predicting phonons, and accuracy on all other properties. Do the authors have an idea of the cause of this phenomenon?

---

> ### Author Response · Authors · 2025-11-24
>
> > After claiming that MD is slow, the authors propose the BSCT test, which requires first-principle evaluations. Unfortunately, I do not believe the BSCT test is very practical in its current state.
> > How would an architecture developer or a model creator use their BSCT in practice if their DFT settings are not consistent with those used in this work and/or if they are interested in applications other than molecules (materials, catalysis, etc.)?
>
> We would suggest the developer to iterate their design on SPICE MACE-OFF and the current BSCT test first, and then train on the target dataset with the identified smooth design: this is similar to the way that we carried out the MPTrj experiments with the smoothness-oriented designs identified by the SPICE ablations.
>
> We are also planning to generate BSCT evaluations for the OMol25 dataset, which will allow larger-scale training (we are contacting the authors of the OMol25 dataset and plan to work with them to construct BSCT). OMol25 is currently the most comprehensive molecular dataset to date, with over 100M+ molecular structures. Models trained on this dataset, if they do well on the BSCT evaluation metrics (among the other metrics in the paper), would serve as an especially reliable starting point for molecular simulations. For materials and catalysis, it remains unclear how BSCT and FSD can generalize, since the notion of bond is not well-defined for material systems. The closest to this is what we have mentioned above, i.e., use the smoothness design choices developed for molecular datasets, and some of this may transfer over to materials. We note that molecular space already covers a vast range of application areas.
>
> > On lines 390-391, the authors mention that "models with proper normalization never output large forces". Is this not worrying, considering that forces can be unbounded in practice? The same concern goes for the energy head of these models.
>
> The argument for never outputting large forces is only applicable to direct force models.  This is because modern machine learning models are usually guaranteed to be normalized in the forward pass (by LayerNorm), while it is generally not guaranteed for the backward pass. Therefore, for gradient-based predictions, forces can be generally unbounded, and this is an argument for why we should use gradient-based MLIPs. On the other hand, indeed energy should also be unbounded in practice, especially when two atoms are colliding, where from first principle we expect a strong inverse power-law repulsive interaction. Indeed, we see in BSCT 1-D energy curves that most MLIPs are not able to learn this inverse power-law repulsive force; the energy heads are not able to predict energy that is 10eV higher than the minimum energy, resulting in a cutoff of the potential energy curve on the compression end. In most settings, this is acceptable since 10eV is significantly higher than the bonding energy, and the bonded atoms would already have escaped the potential well before colliding into another atom. There are two potential ways of addressing this issue (if desired). First, the model can be implemented in GemNet-style, where no normalization is applied in the forward pass, and the model is carefully initialized to give bounded output at random initialization. Second, we can learn the MLIP in a delta-learning fashion, i.e., train an MLIP to learn the difference between DFT labels and a classical force field. If the classical force field is good to an additive epsilon error, then we are guaranteed that the MLIPs will never need to output unbounded energy predictions. These are out of our scope of study in this project, but we can discuss this in more detail in the manuscript.

---

> ### Author Response · Authors · 2025-11-24
>
> > The results in Table 4 seem to indicate a trade-off between smoothness, which is beneficial especially when predicting phonons, and accuracy on all other properties. Do the authors have an idea of the cause of this phenomenon?
>
> Phonon mode predictions are very sensitive to small fluctuations in the PES. Therefore extreme smoothing and aggressive early stopping must be applied to get a good $\kappa_{SRME}$ in practice. These are contradictory to the designs that can yield good energy prediction accuracy, where we want the model to be converged as tightly as possible (which, causes some fluctuations). However, we should note that such fluctuations can be corrected post-hoc. As pointed by Fu et al. [1], increasing the finite differences step size used in phonon mode calculations can improve the phonon mode prediction accuracy. Such improvements are often more significant for unrestricted architectures due to their nature of being more expressive and thus covering more functional space that contains more fluctuations. In our experiments, we see that increasing the finite difference from 0.03Å to 0.1Å can dramatically improve $\kappa_{SRME}$ from 0.69 to 0.34. However, to keep the table presented in the paper a fair comparison to other works, we kept the 0.03Å that was used for calculating the results for other MLIPs.
>
> > the authors claim "we focus on bonds because their ground truth PES is intrinsically smooth". This is incorrect.
>
> We would like to emphasize that the smoothness definition used in our work is for a PES to have no artificial extrema and inflection points according to the definition of chemical smoothness presented in [2]. And per this definition, apart from one minimum and one inflection point that should generally exist for Lennard-Jones-like potential, DFT calculations for bond dissociation are generally free of artificial extrema and inflection points. Furthermore, even taking the definition of smoothness being infinitely differentiable, bond dissociation is still generally smoother than other potential 1-D pathways that can cover far-from-equilibrium structures. One potential alternative could be a 1-D perturbation along a reaction pathway. However, a pathway connecting reactant and product minima may not necessarily be smooth - particularly when the pathway is still in the process of being optimized (as is the case for the majority of the Transition1x data, given that it is constructed of snapshots from NEB path optimizations). Further, while the optimized reaction pathway should be fairly smooth, we lack an intuitive understanding of how the curve should behave - i.e. the path is much more likely to have a rapidly varying second derivative.
>
> > In Figure 5, it seems that the simulations have not equilibrated yet by the time the MD trajectory begins
>
> First, we would like to apologize for the typo made in the figure axis; instead of 100 ps time span, it should be 10ps, which would match with line 347. Second, the design of this experiment is precisely to simulate the process of a ground-state molecule equilibrating with a high-temperature heat bath and to detect if any unrealistic jump in kinetic energy is seen during the process. Such unrealistic jumps are unlikely to be a result of the thermostat but instead due to unbounded forces that originated from the PES non-smoothness.
>
> > Line 392: "confirming non-conservative force field". The forces are conservative in this case since they are calculated by backpropagation
>
> Even in the case of an infinitesimal step size limit, as long as the backpropagation does not consider the delta-function forces arising from discontinuity in energy, the force field would be non-conservative. Generally, delta-function forces are not possible on finite-precision machines, and torch does not have corresponding autograd functionalities implemented for discrete operations such as the top-k used in kNN graphs. Therefore, in any practical scenario, a kNN-graph-based MLIP without proper treatment of Diff-kNN would not conserve energy.
>
>
> [1] Fu, Xiang, et al. "Learning smooth and expressive interatomic potentials for physical property prediction." arXiv preprint arXiv:2502.12147 (2025).
>
> [2] Subotnik, Joseph E., Alex Sodt, and Martin Head-Gordon. "The limits of local correlation theory: Electronic delocalization and chemically smooth potential energy surfaces." The Journal of chemical physics 128.3 (2008).

---

### Official Review · Reviewer_PWxi · 2025-10-31

**Soundness:** 3
**Presentation:** 2
**Contribution:** 3
**Rating:** 6
**Confidence:** 4

**Summary:**

This work introduces a new benchmark for evaluating potential energy surface (PES) smoothness in machine-learning interatomic potentials (MLIPs). The proposed Bond-Smoothness Characterization Test (BSCT) probes one-dimensional bond stretching and compression trajectories, and reports a Force Smoothness Deviation (FSD) score intended to capture spurious extrema or non-smooth behavior. The authors further apply BSCT as a development signal for designing a general-purpose transformer-inspired MLIP architecture with minimal built-in geometric constraints, incorporating smoothness-oriented design choices such as broadened smearing and temperature-controlled attention. The resulting model achieves competitive performance on SPICE, MPTraj, and the Matbench Discovery benchmark suite, while maintaining good smoothness metrics.

Note: An LLM (ChatGPT 5) was used to prepare this review -- it expanded notes into the full review, and suggested some changes in phrasing. It did not provide useful suggestions for the content of the review.

**Strengths:**

- Introduces a physically motivated smoothness benchmark and a corresponding scalar metric (FSD) that quantifies anomalies during controlled bond stretching and compression.
- Constructs a dataset tailored to probe PES smoothness in both near- and far-from-equilibrium regimes.
- Proposes a new minimally constrained transformer architecture, and systematically studies architectural and regularization decisions with respect to PES smoothness.
- Demonstrates strong empirical performance across molecular and materials benchmarks, and attempts to connect physical soundness with downstream stability.

**Weaknesses:**

- The narrative feels divided between presenting a benchmark and proposing a new architecture; a clearer separation of these contributions, and a stronger articulation of how the benchmark concretely guides model design, would improve clarity.
- While the results suggest that FSD correlates with MD stability, broader and more quantitative evidence would help substantiate the claim of predictiveness.
- The paper would benefit from explicit examples where BSCT led to specific architectural choices, especially in cases where intuition or standard smoothness regularization alone would not have suggested the same outcome.
- Physical priors such as rotational equivariance are discussed only briefly; a more explicit comparison with existing physics-informed approaches would strengthen the positioning.

**Questions:**

- Please clarify how BSCT was used to guide architecture decisions in practice. Could the same modifications have been derived by simply optimizing for smoothness via standard regularization techniques? Are there cases where BSCT led to counterintuitive or unexpected design decisions? It would also be interesting to discuss whether automated architecture search using FSD as an objective might reproduce similar choices.
- The paper asserts that FSD correlates with MD stability. Figure 5 is compelling, but a quantitative measure of correlation would make the claim stronger.
- Table 1 highlights that GemNet-T achieves low FSD but still exhibits large energy drift, and similarly the prediction head with the lowest FSD has the worst energy conservation. I assume this reflects non-conservativity rather than smoothness. Please discuss this distinction clearly, as it may otherwise lead to confusion about FSD’s scope and whether this constitutes a limitation of the metric.
- Fu et al. (ICML 2025; https://openreview.net/forum?id=R0PBjxIbgm) in the eSEN model also emphasize smoothness. Please discuss the connection to that work and, if possible, report eSEN’s performance under BSCT to contextualize your metric.
- In Equation (3), the notation could be clearer; explicitly indicating that the derivative is taken with respect to α and evaluated at α would help readability.
- How does FSD compare to measures of non-conservativity such as the Jacobian asymmetry metric introduced by Bigi et al.? A brief, potentially empirical, comparison would help clarfiy what they capture.
- How do the authors define the term "bond"? What's your operational definition?

Minor suggestions:

- The legend of Table 5 states “Number of Steps Per Day,” which likely is intended to mean “Millions of Steps per Day.”
- The proposed architecture shares similarities with the point-edge transformer (Pozdnyakov & Ceriotti, NeurIPS 2023). It would be appropriate to acknowledge this connection.

---

> ### Author Response · Authors · 2025-11-24
>
> > Please clarify how BSCT was used to guide architecture decisions in practice. Could the same modifications have been derived by simply optimizing for smoothness via standard regularization techniques? Are there cases where BSCT led to counterintuitive or unexpected design decisions? It would also be interesting to discuss whether automated architecture search using FSD as an objective might reproduce similar choices.
>
> We would like to answer these questions with the example of how we came up with the temperature-controlled attention design. In the paper, we motivated the temperature-controlled attention by finding that, together with weight decay, temperature can regularize the attention weights. However, in the process of working on this project, it was first proposed as a result of directly inspecting the BSCT evaluation outputs. We plotted out the 1-D curve of energy and force norm due to perturbation of each system in the BSCT dataset, and by inspecting the curves, we noticed that the vanilla model has sharp spikes in force norm due to perturbation near the bond length of 2Å for many systems. After a closer inspection of all components in the model by evaluating on systems near that bond length, we noticed that the attention weights changed abruptly. Therefore, we realized that we need to regularize the attention weights, and proposed two solutions: normalized attention on a sphere or temperature-controlled attention. And with more ablations, we found that the latter can balance accuracy and smoothness better.
> Therefore, our answer is yes, all the designs we proposed can be derived from regularization principles. However, it is often too obscure and difficult for developers to enumerate all potential factors that could compromise smoothness, and BSCT can serve as a great interactive debugging tool that can help find where the regularizations are needed.
>
> >The paper asserts that FSD correlates with MD stability. Figure 5 is compelling, but a quantitative measure of correlation would make the claim stronger.
>
> We thank the reviewer for the comment. We provide a more quantitative and statistically significant version of the experiment we described in Figure 5. We repeat the procedure described in Section 5.2 for 10 different seeds, resulting in 70 distinct trajectories per model per temperature. We calculate the “Max Jump,” defined as the maximum energy change in a 10-fs window. The max jump values are averaged over the 70 trajectories. The values are reported as follows:
>
> |name|FSD (1/Å)|Max Jump 2000K|Max Jump 3000K|Max Jump 5000K|
> |-|-|-|-|-|
> |Vanilla MindScAIP|97.4|9734|813|597|
> |w/ Weight Decay|76.3|1904|681|509|
> |w/ Smear. & Temp.|43.2|490|614|514|
>
> We see a strong correlation between maximum jump and FSD values. For higher temperatures, the covalent bonds dissociate faster and thus reduce the chance for the simulated trajectory to encounter non-smoothness, since the molecule is broken down into pieces in the early stage of the simulation.

---

> ### Author Response · Authors · 2025-11-24
>
> > Table 1 highlights that GemNet-T achieves low FSD but still exhibits large energy drift, and similarly the prediction head with the lowest FSD has the worst energy conservation. I assume this reflects non-conservativity rather than smoothness. Please discuss this distinction clearly, as it may otherwise lead to confusion about FSD’s scope and whether this constitutes a limitation of the metric.
>
> > How does FSD compare to measures of non-conservativity such as the Jacobian asymmetry metric introduced by Bigi et al.? A brief, potentially empirical, comparison would help clarfiy what they capture.
>
> We answer these two questions together as they are connected.
>
> We emphasize that a low FSD does not imply energy conservation and vice versa. We can construct counterexamples for that: by adding a constant force vector to the ground-truth force field, FSD stays unchanged while violating conservativity. On the other hand, a constant-zero force field is conservative while having high FSD. In fact, achieving low FSD is generally harder for a conservative MLIP than a direct force MLIP. This is because modern machine learning models are usually guaranteed to be normalized in the forward pass (by LayerNorm), while it is generally not guaranteed for the backward pass. Therefore, it is a harder task for MLIPs to output bounded forces when predicting with autograd. As a result, the three dimensions that we benchmarked our models, in-distribution accuracy, conservativeness, and BSCT FSD, should be considered as distinct and necessary conditions (instead of sufficient) for a model to be physically sound. In our case, we measured the Jacobian asymmetry metric of the MinDScAIP models with Diff-kNN, and they are all close to zero up to machine precision.
>
> Meanwhile, we note that GemNet-T is, in theory, a conservative model. The non-conservative behaviour seen in the paper is most likely a result of being tested on out-of-distribution structures. We followed the procedure proposed by Fu et al. in the eSEN work to test GemNet-T trained on SPICE MACE-OFF on MD22 structures. These structures are out-of-distribution relative to SPICE, and GemNet-T could predict a rapidly varying PES that the finite timestep integrator cannot bound the error, and thus causes an energy drift in microcanonical simulation.
>
> > Fu et al. (ICML 2025; https://openreview.net/forum?id=R0PBjxIbgm) in the eSEN model also emphasize smoothness. Please discuss the connection to that work and, if possible, report eSEN’s performance under BSCT to contextualize your metric.
>
> In Fu et al. [1], the operational definition of smoothness is for the PES to have bounded derivatives at a magnitude that is similar to the ab initio method, and they measure these by running long microcanonical simulations. In essence, this is closer to the mathematical definition of a smooth function (admitting arbitrary order derivatives). Meanwhile, our work takes the definition of chemical smoothness (see [2] and Section 3), where a smooth PES is defined as a PES without any artificial extrema and inflection points. To make clear our contribution, we emphasize the following differences/similarities:
> We evaluate the smoothness at far-from-equilibrium configurations, which is not discussed in the eSEN paper.
> Meanwhile, we still studied the smoothness definition that the eSEN paper focused on in our work (we adopted the exact protocol to measure their definition of smoothness). However, to avoid confusion, we did not call it “smoothness,” but instead the test of energy conservation.
> To improve performance, eSEN [1] featurizes radial distances with few number of basis functions to reduce high frequencies, while we increased the basis width to control derivatives. We chose the latter as it can give a stronger theoretical guarantee, as shown in Appendix D.
>
> Unfortunately, we are unable to report eSEN’s performance under BSCT since there are no public weights for eSEN trained on SPICE MACE-OFF, and training eSEN from scratch is not practical due to computational resource constraints.
>
> [1] Fu, Xiang, et al. "Learning smooth and expressive interatomic potentials for physical property prediction." arXiv preprint arXiv:2502.12147 (2025).
> [2] Subotnik, Joseph E., Alex Sodt, and Martin Head-Gordon. "The limits of local correlation theory: Electronic delocalization and chemically smooth potential energy surfaces." The Journal of chemical physics 128.3 (2008).

---

> ### Author Response · Authors · 2025-11-24
>
> > In Equation (3), the notation could be clearer; explicitly indicating that the derivative is taken with respect to α and evaluated at α would help readability.
>
> > The legend of Table 5 states “Number of Steps Per Day,” which likely is intended to mean “Millions of Steps per Day.”
>
> > The proposed architecture shares similarities with the point-edge transformer (Pozdnyakov & Ceriotti, NeurIPS 2023). It would be appropriate to acknowledge this connection.
>
> We thank the reviewer for the suggestions. We will incorporate these into the revised manuscript.
>
> >How do the authors define the term "bond"? What's your operational definition?
>
> Our operational definition of a chemical bond is to use the SMILE string provided by the SPICE dataset. In practice, it may be hard to define bonds for an arbitrary molecular structure. However, in the case of SPICE and many other molecular DFT datasets, the structures are generated by running classical MD of a known molecule, where, if the temperature is not too high, the atoms will stay covalently bonded, and we can take the bonds defined by the initial structure (SMILE string).

---

> > ### Comment · Reviewer_PWxi · 2025-11-26
> >
> > I thank the authors for their timely and thorough reply -- I have no further questions for now. I trust that the authors will revise their paper in line with our discussion, and will raise my score in anticipation. Thanks!

---

### Official Review · Reviewer_dHUT · 2025-11-01

**Soundness:** 3
**Presentation:** 3
**Contribution:** 2
**Rating:** 4
**Confidence:** 4

**Summary:**

This paper proposes the Bond Smoothness Characterization Test (BSCT) - a low-cost test to directly assess potential-energy-surface (PES) smoothness for ML interatomic potentials by performing 1-D bond stretches/compressions.

Then the paper incorporates three types of inductive biases aimed to promote smoothness, differentiable neighborhood construction, temperature-controlled attention, and broadened radial smearing. The paper finds that the BSCT test is capable of indicating the associated increase of smoothness, and can serve as a cheaper alternative to molecular dynamics to estimate the stability of the model.

**Strengths:**

The paper argues convincingly that accuracy is not the same as reliability and that PES smoothness matters for practical simulations. This point is often overlooked, and many current models are mostly focused on the errors in energies and forces that do not provide a reliable estimation of the quality of the interatomic potential.

The proposed smoothness indicator is cheap and was found to be able to predict instabilities in much more costly MD experiments.

The three proposed strategies were found to be able to promote smoothness in practice.

**Weaknesses:**

The proposed metric, Bond Smoothness Characterization Test, is based on an isolated and, I would say, artificial part of the Potential Energy Surface. The forces are evaluated for configurations that are hardly expected to originate during molecular dynamics simulations - a molecule is stretched across one of the bonds, while the two parts are undisturbed.

The non-smoothness of PES is expected to affect the stability of MD if it is present for the part of the configurational space that is accessible and often visited during MD trajectories. The chosen part of configurational space to judge the overall smoothness or non-smoothness of PET doesn't seem to be especially important for MD, or at least more important than the other, random parts of PES. There are no ablations comparing the proposed smoothness metric to alternatives.

Additionally, the claim that the FSD values can be used to assess the stability of MD is not fully supported. Currently, it is given by Figure 5, which shows this dependency only for one particular case. It is unclear if this is just a coincidence or a rule, and it would be nice to have some statistically significant verification.

**Questions:**

Why do the authors think that this particular choice of the smoothness criteria is more relevant for MD stability compared to others?

Can the authors propose other alternatives, such as simply estimating the Hessian of PET at random off-equilibrium points or others, and show that the proposed metric is better (e.g., better predicts stability/instability of MD) compared to other metrics?

Can the authors provide a statistically significant justification that the FSD indeed predicts well MD stability?

---

> ### Author Response · Authors · 2025-11-24
>
> > Why do the authors think that this particular choice of the smoothness criteria is more relevant for MD stability compared to others?
>
> > The chosen part of configurational space to judge the overall smoothness or non-smoothness of PET doesn't seem to be especially important for MD
>
> Given our interest in assessing MLIP smoothness in far-from-equilibrium configurations, as there are many applications that critically depend on far-from-equilibrium predictions, directly comparing them to DFT on single-point calculations is not practical. This is because the MLIP energy and forces error do not always correlate with their performance in MD or other downstream physics applications. To more comprehensively evaluate how MLIPs behave, we need to analyze their qualitative performance along a continuous grid. While it is possible to construct high-dimensional grids in this assessment, we have chosen 1-D paths that can be efficiently evaluated at test-time. Empirically, we also found that the smoothness of 1-D paths is already a hard task for MLIPs, making them a good proxy for optimizing the overall smoothness of the PES. By concentrating on these 1-D evaluations, we are able to streamline the testing process while still capturing essential characteristics of the MLIPs’ performance.
>
> We chose to test on 1-D perturbation of chemical bonds since they can be efficiently evaluated and have a strong theoretical guarantee to be smooth. On the other hand, other 1-D pathways, such as reaction pathways, may contain rapidly varying second derivatives, making it harder to isolate erroneous non-smoothness from the MLIP, which is quite straightforward in the bond stretching/compression context, where we know the PES should be very smooth. Moreover, though the breakage of an isolated bond may be “unphysical,” we believe that the smoothness of the PES region governing a more realistic reaction involving simultaneous bond breakage and formation is strongly correlated with the PES smoothness under isolated bond breakage - both are far from equilibrium and involve complex interactions in the intermediate regime between covalent and non-covalent.
>
> Therefore, in response to the reviewer’s comment that BSCT “doesn't seem to be especially important for MD”, we would like to stress that (1) we chose to test bonds because they allow easier isolation of erroneous behaviour of MLIPs, not because they are more important than other parts of the PES and (2) there are few alternatives to BSCT other than long MD simulations that would allow one to assess far-from-equilibrium PES smoothness, and to our knowledge, BSCT is the first-of-its-kind metric proposed to explicitly evaluate this.
>
> >Can the authors propose other alternatives, such as simply estimating the Hessian of PET at random off-equilibrium points or others, and show that the proposed metric is better (e.g., better predicts stability/instability of MD) compared to other metrics?
>
> We thank the reviewer for the suggestion. However, we note that constructing such a Hessian dataset with DFT labelling requires a significant amount of computational resources: O(100) single-point evaluations for one Hessian evaluation, and O(10^3) of such Hessians to get statistically significant results. This is unfortunately out of our computational capabilities. Other alternatives available in the literature are mostly MD-based. Some non-MD-based approaches include torsion scans (e.g., TorsionNet-500 [1]). However, these metrics only evaluate near-equilibrium, which is different from the target configuration space that we would like to probe, which includes both near-equilibrium and far-from-equilibrium accuracy.
>
> > Can the authors provide a statistically significant justification that the FSD indeed predicts well MD stability?
>
> We thank the reviewer for the comment. We provide a more quantitative and statistically significant version of the experiment we described in Figure 5. We repeat the procedure described in Section 5.2 for 10 different seeds, resulting in 70 distinct trajectories per model per temperature. We calculate the “Max Jump,” defined as the maximum energy change in a 10-fs window. The max jump values are averaged over the 70 trajectories. The values are reported as follows:
>
> |name|FSD (1/Å)|Max Jump 2000K|Max Jump 3000K|Max Jump 5000K|
> |-|-|-|-|-|
> |Vanilla MindScAIP|97.4|9734|813|597|
> |w/ Weight Decay|76.3|1904|681|509|
> |w/ Smear. & Temp.|43.2|490|614|514|
>
> We see a strong correlation between maximum jump and FSD values. For higher temperatures, the covalent bonds dissociate faster and thus reduce the chance for the simulated trajectory to encounter non-smoothness, since the molecule is broken down into pieces in the early stage of the simulation.
>
> [1] Rai, Brajesh K., et al. "Torsionnet: A deep neural network to rapidly predict small-molecule torsional energy profiles with the accuracy of quantum mechanics." Journal of Chemical Information and Modeling 62.4 (2022): 785-800.

---

### Author Response · Authors · 2025-11-24
**General Response**

We thank the reviewers for the comments and the time taken to review the paper. We would like to use this response to clarify some questions that the reviewers commonly brought up:

> The evidence of FSD as an early proxy for MD stability.

We provide a more quantitative and statistically significant version of the experiment we described in Figure 5. We repeat the procedure described in Section 5.2 for 10 different seeds, resulting in 70 distinct trajectories per model per temperature. We calculate the “Max Jump,” defined as the maximum energy change in a 10-fs window. The max jump values are averaged over the 70 trajectories. The values are reported as follows:

|name|FSD (1/Å)|Max Jump 2000K|Max Jump 3000K|Max Jump 5000K|
|-|-|-|-|-|
|Vanilla MindScAIP|97.4|9734|813|597|
|w/ Weight Decay|76.3|1904|681|509|
|w/ Smear. & Temp.|43.2|490|614|514|

We see a strong correlation between maximum jump and FSD values. For higher temperatures, the covalent bonds dissociate faster and thus reduce the chance for the simulated trajectory to encounter non-smoothness, since the molecule is broken down into pieces in the early stage of the simulation.

> Justification for the particular choice of bond perturbation.

We emphasize key reasons for us to use bond perturbation:
There are very few alternatives to a BSCT-style 1-D perturbation other than long MD simulations that would allow one to assess far-from-equilibrium PES smoothness. Testing chemical smoothness requires comparing the qualitative behaviour of an MLIP at neighboring points, as single-point error does not fully correlate to actual downstream physics applications. A 1-D grid is the minimal configuration that we can efficiently test on.
We chose to test bonds because they allow easier isolation of erroneous behaviour of MLIPs, not because they are more important than other parts of the PES. A 1-D path corresponding to a bond has the strongest theoretical guarantee to have a smooth potential energy curve, which allows for better isolation of MLIP’s behaviour compared with other alternatives, such as reaction pathways that can have rapidly varying derivatives.
To our knowledge, BSCT is the first-of-its-kind metric proposed to explicitly evaluate far-from-equilibrium smoothness.

> How does the work relate to the existing work in the field that studies smoothness.

Most of the existing work focuses on smoothness as the quality of having bounded derivatives. In Fu et al. [1], the operational definition of smoothness is for the PES to have bounded derivatives at a magnitude that is similar to the ab initio method, and they measure these by running long microcanonical simulations. In essence, this is closer to the mathematical definition of a smooth function (admitting arbitrary order derivatives). Meanwhile, our work looks at evaluating the definition of chemical smoothness (see [2] and Section 3), where a smooth PES is defined as a PES without any artificial extrema and inflection points.



[1] Fu, Xiang, et al. "Learning smooth and expressive interatomic potentials for physical property prediction." arXiv preprint arXiv:2502.12147 (2025).

[2] Subotnik, Joseph E., Alex Sodt, and Martin Head-Gordon. "The limits of local correlation theory: Electronic delocalization and chemically smooth potential energy surfaces." The Journal of chemical physics 128.3 (2008).

---

### Author Response · Authors · 2025-12-03
**Final Remarks to the AC**

We thank all reviewers and ACs for their service. As our final remarks, we would like to emphasize a few points that were discussed during the rebuttal and discussion phase:

1. **The evidence of our proposed smoothness evaluation metric, Force Smoothness Deviation, as an early proxy for molecular dynamics stability**: We expanded on the experiment described in Section 5.2, providing more quantitative and statistically significant evidence for the correlation between FSD and MD stability. We now report results over 10 random seeds (70 trajectories per model per temperature), showing a consistent and strong correlation between FSD and maximum energy jumps across temperatures.

2. **Rationale, scope, and limitations of our smoothness test and dataset, the Bond Smoothness and Characterization Test (BSCT)**: We addressed the reviewers’ concerns by explaining why the 1-D bond perturbations for this test are a useful and theoretically grounded probe of chemically smooth potential energy surface (PES) behavior, while also emphasizing that BSCT is not intended as the only relevant metric. We clarified the distinction between chemical smoothness and the mathematical notion of smoothness; elaborated on why bond perturbations offer a clean, interpretable benchmark for isolating MLIP-induced non-smoothness. We also discussed alternatives (e.g., Hessian probing, reaction-path scans, metrics from running MD simulation) and why they are either computationally prohibitive or less suitable for systematic far-from-equilibrium PES smoothness evaluation.

3. **Clarifications on model design, the role of BSCT in guiding architecture choices, and relations to prior work**: The reviewers asked how BSCT concretely influenced architectural decisions. We provided a clear explanation illustrating how temperature-controlled attention emerged directly from diagnosing BSCT force-curve anomalies. We further clarified the distinction between smoothness, in-distribution accuracy, and conservativity, and discussed how FSD relates to metrics such as Jacobian asymmetry. We added discussion of connections to eSEN (Fu et al.), point-edge transformers (Pozdnyakov & Ceriotti), and other relevant prior architectures.

We incorporated these important points raised during the discussion phase, including the expanded experiments, rationale for the BSCT procedure, and flow of narrative to our revised manuscript. We want to highlight that, following the discussion, reviewer PWxi appreciated our clarifications and raised the score.

Finally, we would like to reiterate our key contributions:

-  We proposed a novel metric, the force smoothness deviation (FSD), to efficiently detect the non-smoothness in ML interatomic potentials built the bond smoothness and characterization test (BSCT) dataset to support it
-  guided by BSCT, we explored different model architectural designs that optimize the physical soundness while preserving the scalability and expressiveness of the architecture
- We designed the Diff-kNN algorithm, enabling both the efficiency of the k-Nearest-Neighbor graph and the differentiability that is crucial to enforcing energy conservation. Note that on-conservative models also benefit from using Diff-kNN since they provide a smoother cutoff than kNN graphs.

We consider these contributions to have long-term impacts in the study of MLIPs.

---

### Meta-Review · Area_Chair_ZfJi · 2026-01-08

**Summary:**

The paper introduces a metric BSCT for assessing the smoothness of MLIPs, and proposes a new architecture MinDScAIP by searching a few design choices under the view of this metric and other conventional metrics. Reviewers acknowledge the emphasis on measuring and designing for smoothness beyond just prediction accuracy, the ambition to detect ``discontinuities, artificial minima, and spuriously large forces'', the systematic architectural investigation, and the advantageous empirical results.

Reviewers also expressed concerns including:
1. Seemingly limited representativeness of BSCT for PES smoothness, as it only probes a very specific slice of structural space, and it may not even be the most relevant region for typical MD trajectories.
2. Indirect evaluation for running real workloads. While the authors showed that the BSCT metric indicates artifacts in a predefined MD trajectory, a more relevant evaluation to directly running MD is still missing.
3. The connection between the smoothness metric and model architecture design is not sufficiently intimate, in that the smoothness study does not directly lead to novel design spaces.
4. A clearer contextualization relative to existing smoothness-focused models and physics-informed or equivariant architectures are expected, as well as more explicit discussion of similarities and differences.

**Reviewer Concerns:**

After reading authors' rebuttal and additional messages,
1. I do not quite satisfied with authors explanation. The authors mentioned bond breaking against reaction, but bond breaking is often a part of reaction process, which may also lead to violate change of energy. Regarding relevance, I would perceive the change along tortional angle more relevant, and bond angle would also be in a similar role as the bond length if considering far off-equilibrium structures. Although the authors claimed the desire to detect artificial minimum, I would perceive the proposed approach very limited for this target. On periodic systems, a spurious local minimum may not be just a bond length shift but would be e.g., occupying a different spatial group or demonstrating a shear. I would neither treat the argument of using OMol25 dataset for first-principled labels along bond compressing/stretching as a decent and principled solution.
2. Although the authors provided more experimental results showing the correlation between BSCT and MD artifact, the presented result only probes a specific aspect of MD stability. I would expect more concerned quantities in practice to monitor, and more importantly, the used MD trajectory is not by the model itself.
3. The authors listed an example of modification choice inspired by the metric. I would treat it as acceptable.
4. Discussions on the relation to other metrics for smoothness and model designs for smoothness look fine to me. Nevertheless, I would treat the proposed metric and model as another choice and I feel no particularly clear advantage. Moreover, I would expect inspections on recent smoothness-driven architectures under BSCT.

Moreover, the authors relieved the equivariance guarantee in designing the architecture. The investigations do not reflect this but this choice may also pose potential risks of e.g., skewed sample distribution that may bias MD simulation statistics. A more careful study is expected. Moreover, in the Diff-kNN part, I'm confused if a strict radius cutoff (i.e., treat the weight in Eq. (8) lower than a threshold as zero) is conducted. If yes, then I cannot quite imagine why this is still smooth, especially considering that conventional spatial cutoff could employ a smooth envelope to guarantee smoothness which is not the case of Eq. (8). If not, then this solution of smoothness would be trivial: it just operates on a fully connected graph.

With these considerations, I tend to recommend a reject to this submission.

**Reviewer Scores:**

I feel no particular reason that could motivate reviewers to raise their score. (Comments from Reviewer EEZ4 and A4yb are even leaning towards a negative rating.)

---

### Decision · Program_Chairs · 2026-01-26

Reject